# Has COVID-19 Affected DTP3 Vaccination in the Americas?

**DOI:** 10.3390/vaccines12030238

**Published:** 2024-02-25

**Authors:** Ines Aguinaga-Ontoso, Sara Guillén-Aguinaga, Laura Guillén-Aguinaga, Rosa Alas-Brun, Enrique Aguinaga-Ontoso, Esperanza Rayón-Valpuesta, Francisco Guillén-Grima

**Affiliations:** 1Department of Health Sciences, Public University of Navarra, 31008 Pamplona, Spain; sguillen.4@alumni.unav.es (S.G.-A.); guillen.124514@e.unavarra.es (L.G.-A.); rosamaria.alas@unavarra.es (R.A.-B.); 2Healthcare Research Institute of Navarra (IdiSNA), 31008 Pamplona, Spain; 3San Juan Primary Health Carece Center, Navarra Health Service, 31008 Pamplona, Spain; 4Department of Nursing, Kystad Helse-og Velferdssenter, 7026 Trondheim, Norway; 5Department of Sociosanitary Sciences, University of Murcia, 30120 Murcia, Spain; aguinaga@um.es; 6Department of Nursing, Complutense University of Madrid, 28040 Madrid, Spain; erayon@ucm.es; 7Department of Preventive Medicine, Clínica Universidad de Navarra, 31008 Pamplona, Spain; 8CIBER in Epidemiology and Public Health (CIBERESP), Institute of Health Carlos III, 46980 Madrid, Spain

**Keywords:** DTP vaccine, America, COVID-19, vaccine coverage, Joinpoint regression, healthcare system, vaccination rates, trends, segmented regression

## Abstract

Background: In the Americas, deaths by diseases avoidable with vaccines are a significant contributor to child mortality. An essential means of reducing this is through broad vaccine coverage. The COVID-19 pandemic has posed a potential disruption to vaccine coverage due to its effects on the healthcare system. Objectives: this study aims to evaluate the impact of the COVID-19 pandemic on DTP3 vaccination coverage in the Americas, investigating trends from 2012 to 2022 to identify significant changes, regional disparities, and the overall effect of the pandemic on progress towards global immunization targets. Methods: This study used the coverage data for the third dose of the diphtheria, tetanus, and pertussis vaccine (DTP3) pulled from UNICEF databases spanning 2012 to 2022. We conducted a Joinpoint regression to identify points of significant trend changes. The annual percentage change (APC) and 95% confidence intervals (95% CIs) were calculated for America and its regions. We also used segmented regression analysis. Using the Chi-square test, we compared DTP3 vaccination coverage for each country between 2019 and 2022. Results: Overall, America saw a decrease in vaccine coverage during this period, with an APC of −1.4 (95% CI −1.8; −1.0). This trend varied across regions. In North America, the decrease was negligible (−0.1% APC). South America showed the steepest decrease, with an APC of −2.5%. Central America also declined, with an APC of −1.3%. Our findings suggest a concerning trend of declining DTP-vaccination rates in the Americas, exacerbated in certain regions, in the wake of the COVID-19 pandemic. The absolute decrease in vaccine coverage in the Americas was −4% between 2019 and 2022, with the most important drop being in Central America (−7%). However, six countries reported increased vaccination rates post-COVID-19, led by Brazil, with a 7% increase. Conversely, twenty-two countries registered a decline in DTP3 vaccine coverage, with the average decrease being −7.37%. This decline poses an important challenge to achieving the WHO’s target of 90% coverage for the third dose of DTP by 2030, as evidenced by the reduction in the number of countries meeting this target from 2019 to 2022. Conclusions: The COVID-19 pandemic has impacted vaccine coverage in America, leading to a decrease, especially across Central America.

## 1. Introduction

Immunization is a cornerstone in public health, pivotal in diminishing the prevalence and severity of infectious diseases and their effects on health outcomes [1,2,3]. Vaccine-preventable diseases remain a significant public health concern globally. Diphtheria, tetanus, and pertussis (DTP) vaccines prevent these life-threatening diseases, especially among children. The DTP vaccine’s broad coverage is imperative for public health protection. However, the COVID-19 pandemic, emerging in early 2020, posed significant challenges to healthcare systems globally [4], potentially impacting routine immunization services, including DTP vaccination. Programs emphasizing immunization, especially those incorporating the DTP vaccine, are central to reducing the worldwide incidence of diseases preventable by vaccines, thereby improving the lives of countless individuals [5,6,7,8]. The World Health Organization (WHO) launched, in response to this global health challenge, the Global Vaccine Action Plan (GVAP) for the period 2011–2020, initiated in 2012, to achieve widespread routine immunization for children across the globe [9].

Child mortality due to vaccine-preventable diseases is a significant concern, with pertussis alone accounting for over 300,000 deaths annually [10]. Factors influencing disparities in vaccination rates include geographic location (urban vs. rural), socioeconomic status, educational level [11,12], and the regularity of maternal prenatal care [13]. Parents in rural areas believe vaccine-preventable diseases (VPD) are not severe enough to justify vaccination [14]. The emergence of the COVID-19 pandemic has placed an extraordinary burden on global healthcare infrastructures, potentially impacting the continuity of standard immunization efforts [15,16,17,18,19,20]. Disruptions in the supply chain of vaccines, the redirection of healthcare resources and personnel to COVID-19 management [21], the logistic challenges of COVID-19 immunization [22], and heightened public reluctance to visit medical facilities due to infection risk have all been implicated in the observed decrease in vaccination rates [23,24,25,26,27,28]. Such a downturn in immunization coverage poses a severe danger to millions of children, leaving them vulnerable to diseases preventable by vaccines [29]. The pandemic has increased the likelihood of outbreaks of VPD [12]. Lower–middle-income regions with low vaccine coverage and circulating vaccine-derived viral strains, such as polio, bore an additional burden of zero-dose children. All over the world, there were 18.2 million zero-dose children in 2021 [30] that were more vulnerable to VPDs. [12] A precise understanding of the disruption’s impact is vital for tailoring public health policies and building resilience for future crises [31]. Research has shown a global decline of 7.7% in DTP3 coverage and of 7.9% in MCV1 (first dose of measles-containing vaccine) coverage up to December 2020 [32].

In assessing the success of DTP-vaccination programs, we focus on the proportion of infants receiving DTP3, a standard and widely recognized metric [33]. This indicator reflects the effectiveness of the immunization programs in reaching target demographics and completing the primary vaccination series, providing optimal protection against diphtheria, tetanus, and pertussis [34]. The analysis of both DTP1 and DTP3 coverages is essential, with DTP1 coverage indicating the initial reach and engagement of health services [35,36] and DTP3 demonstrating the success in administering the entire course of the vaccine [37]. For instance, a meta-analysis highlighted a significant dropout rate in vaccinations in Africa, with notable variations between countries [35,36]. Discrepancies between DTP1 and DTP3 coverages can shed light on challenges in patient retention and other barriers such as healthcare access, affordability, and education [37].

The Americas ranks as the second-worst region globally regarding vaccine coverage [38,39,40]. Two countries in America were included in the Immunization Agenda 2030 in 2021: Brazil was ranked seventh and Mexico in 15th. In 2022, Mexico disappeared from the list, and Brazil descended to the eighth position [41].

The pandemic’s disruption had widespread implications for DTP-vaccination coverage. Healthcare resources were reallocated to address the pandemic, leading to the neglect of routine vaccination programs. Social distancing and lockdowns are public health measures that may have hindered access to vaccination services. Moreover, the pandemic could have affected public perception and confidence in vaccines, further challenging vaccination efforts. In the United States, during the pandemic, there was a decrease in pediatric primary care visits [42]. Recent data points to a substantial impact of the COVID-19 pandemic on DTP-vaccination trends [43]. A significant decline in administered doses of DTP-containing vaccines was observed in the first half of 2020. This trend was not limited to specific regions but was a global phenomenon with varying degrees of impact across various parts of the world. In Africa, for instance, there was a notable decrease in DTP3 coverage post-2019 [44].

Similarly, a decline in vaccination coverage for multiple vaccines was observed in Latin America, with catch-up strategies implemented to address missed vaccinations [45]. Reports from WHO and UNICEF indicate a notable global decrease in child vaccinations, with DTP3 coverage falling by 5% between 2019 and 2021 [30]. The CDC’s 2021 data confirms this trend, showing the lowest global DTP3 coverage since 2008 [46]. From 2021 to 2022, there was a notable increase in global vaccination coverage for the first dose of the DTP vaccine, from 86% to 89%, and the measles-containing vaccine, from 81% to 84%. However, these levels did not return to the pre-pandemic coverage rates of 90% and 86%, respectively. Despite the challenges of the pandemic, there have been signs of recovery. By 2022, global DTP immunization coverage nearly returned to pre-pandemic levels, although millions of infants still lacked initial or complete vaccination [35,47]. This recovery highlights the resilience of health systems and the importance of ongoing assessment and the implementation of catch-up vaccination strategies, particularly for vulnerable populations, to ensure vaccine-coverage equity and health system resilience. This recovery in vaccination coverage was uneven across different regions and countries, with slower progress in low-income countries [48]. This data, part of the World Health Assembly’s endorsement of the Immunization Agenda 2030 (IA2030), highlights the ongoing challenge of restoring and improving global vaccination coverage after the COVID-19 pandemic, particularly among low- and lower–middle-income countries [49].

This study aims to analyze the trends in DTP3 vaccination coverage in America from 2012 to 2022, emphasizing the influence of the COVID-19 pandemic. Based on the evidence discussed, we posit that the pandemic has adversely affected vaccination rates [46,49].

Considering the available data, we observe a significant global impact of the COVID-19 pandemic on DTP-vaccination trends. Reports from WHO and UNICEF highlight a worldwide decline in child vaccinations during this period [50]. The risk for children in developing vaccine-preventable diseases is thus elevated. This study aims to assess the impact of the COVID-19 pandemic on DTP-vaccination coverage in the Americas. We hypothesize that the pandemic has led to declining vaccination rates, potentially reversing previous progress toward global immunization targets. This paper examines trends in DTP3-vaccine coverage across the Americas from 2012 to 2022, identifying significant changes and regional disparities considering the pandemic. The findings aim to provide a detailed understanding of the pandemic’s impact on DTP vaccination and guide public health strategies to address these challenges, ensuring continued progress towards global vaccination goals. We will focus primarily on the repercussions of the COVID-19 pandemic under the hypothesis that pandemic-related disruptions have affected vaccination programs [46].

## 2. Materials and Methods

Vaccine coverage data for individual countries were sourced from the United Nations Children’s Fund, in databases from 2012 through 2022 [51]. We omitted the following territories: French Guiana, Greenland, Alaska, Anguilla, Aruba, Bermuda, Bonaire, Curaçao, and Guadeloupe. We also acquired regional estimates from the United Nations Children’s Fund database [51]. The information regarding the annual count of newborns by country was gathered from the United Nations Children’s Fund database [52] and the World Bank [53].

### 2.1. Regional Analysis

Regional aggregated data for North America, Latin America, and the Caribbean were obtained from the United Nations Children’s Fund. Due to the absence of regional data from South America, Central America, and the Caribbean in the United Nations Children’s Fund data, we calculated these figures using birth-weighted vaccination rates. This method of calculating data, based on the births in each country, was essential for deriving accurate estimations of DTP3-vaccination coverage in these regions. The countries included in each region were as follows: in Central America, the countries were the Republic of Guatemala, the Republic of Panama, the Republic of Costa Rica, the Republic of Nicaragua, the Republic of El Salvador, the Republic of Honduras, and Belize. In North America, the countries assessed included the United States of America, the United Mexican States, and Canada. The Caribbean region’s analysis encompassed Jamaica, Saint Lucia, Grenada, the Bahamas, the Republic of Cuba, Saint Vincent and the Grenadines, Barbados, the Republic of Haiti, the Republic of Trinidad and Tobago, the Dominican Republic, the Federation of Saint Kitts and Nevis, Antigua and Barbuda, and the Commonwealth of Dominica. Lastly, in South America, the countries involved were the Federative Republic of Brazil, the Republic of Chile, the Bolivarian Republic of Venezuela, the Republic of Peru, the Argentine Republic, the Republic of Ecuador, the Plurinational State of Bolivia, the Republic of Colombia, Uruguay, Paraguay, Suriname, and Guyana.

### 2.2. Statistical Analysis

We utilized Joinpoint regression, a methodology that has previously been applied in the examination of vaccination trends [44]. We calculated the annual percentage change (APC) to gauge the extent of variation in each trend. In these statistical models, vaccine coverage was the dependent variable, while the year was the independent variable. The models assumed constant variance (homoscedasticity). We conducted a Durbin–Watson test to evaluate the presence of autocorrelation within the time series data [54]. First-order autocorrelation estimated from the data was computed in all the cases.

We employed an interrupted time series analysis method. This technique is considered the most effective quasi-experimental approach for assessing the impact of external events or interventions, such as the COVID-19 pandemic [55,56,57]. Our study analyzed eleven years, eight years pre-COVID (2012–2019) and three years, 2020–2022, post-COVID. The model of the analysis follows the equation below.
DTP3*_t_* = Intercept + *β*_1_Year + *β*_2_COVID-19 + *β*_3_YearCOVID-19 + *ε**_t_*(1)
where “DTP3” is the number of doses administered, Year is the year of the calendar, and COVID-19 is a dummy variable that has value “1” for the pandemic years (2020–2022); “YearCOVID-19” is an interaction between Year and COVID-19.

Furthermore, we compared DTP3-vaccination coverage for each country in 2019 and its respective coverage data in 2022. Our examination of DTP3-vaccination coverage across American countries involved a comparison of coverage rates for 2019 and 2022. A 2 × 2 contingency table was constructed for each country, delineating the number of vaccinated and unvaccinated children based on the total number of births and the reported DTP3-vaccination percentages for each year. These tables facilitated the comparison of vaccination rates over the specified period, allowing for the assessment of the impact of COVID-19. Chi-square tests were employed to evaluate the significance of differences in DTP3-vaccination rates across various American countries between 2019 and 2022 in the statistical analysis of vaccination-rate changes.

### 2.3. Software

All Joinpoint analysis calculations were performed utilizing Joinpoint (Version 5.0.2. May 2023) [58,59]. The statistical comparison of rates between 2019 and 2022 was conducted using the IBM Statistical Package for the Social Sciences, version 27 (IBM Corp., Armonk, NY, USA). The interrupted time series analysis was computed using the “segmented” [60,61,62,63] package in R (version 4.3.1, 16 June 2023 ucrt) [64], under Rstudio (Version 1 September 2023 Build 494) [65]. This was complemented using the ‘ggplot2′ package for advanced graphical representations [66] and the ‘readxl’ package for seamlessly importing Excel data files [67]. We computed post hoc power calculations using the G*Power Software (Version 3.1.9.6) [68,69]. Maps were created with Mapchart (v 4.3.2.) [70]. The range of colors for the maps was elaborated with Colorbrewer (v.2.0) [71,72].

## 3. Results

Between 2012 and 2021 in the Americas, DTP3-vaccination rates displayed an overall APC of −1.4% with a 95% confidence interval (CI) ranging from −1.8 to −1.0 (*p* < 0.001). The Joinpoint analysis identified two distinct periods within this period. From 2012 to 2016, the APC was −0.7% (95% CI: −2.9 to 1.5; *p* = 0.464) during the first period, indicating a slight decrease. In contrast, a marked decline was observed in the second period, from 2016 to 2022, where the APC steepened to −1.8% (95% CI: −2.9 to −0.7), indicating a downward trend. These findings suggest a shift in the trajectory of vaccination rates over the decade, with a notable decline in the latter half of the period (Table 1, Figure 1).

The Joinpoint analysis for regional third-DTP-dose coverage in America from 2012–2022 reveals varied trends across regions. In North America, the overall APC for the total period was −0.1% (95% CI: −0.2 to 0), indicating a negligible decrease in vaccination rates. In Table 2, we can see the Joinpoint analysis for 2014. From 2012–2014, there was a slight increase; from then on, there was a decrease, as depicted in Figure 2.

In contrast, Latin America and the Caribbean exhibited a more pronounced decline in vaccination rates. The total period APC was −2.1% (95% −2.7 to −1.5). Further analysis within this region revealed a Joinpoint in 2016. There were two distinct periods: the first period (2012–2016) showed an APC of −0.9% (95% CI: −4.4, 2.6, *p* = 0.531), indicating a stable trend. However, during the second period (2016–2022), the decline in vaccination rates was more substantial, with an APC of −2.7% (95% CI: −4.5, −0.9, *p* = 0.010), indicating a decrease in vaccination rates, as shown in Figure 3. These findings highlight substantial regional differences in third-DTP-dose-coverage trends within Latin America and the Caribbean, which are visually depicted in Figure 4, Figure 5 and Figure 6, illustrating the third-DTP-dose-vaccination-rate trends in Central America, the Caribbean, and South America, respectively, highlighting the Joinpoints.

During the research period spanning from 2012 to 2022, there were notable variations in the vaccination rates for the third dose of DTP across different areas in Central America, the Caribbean, and South America. In Central America, the overall APC was −1.3% (95% CI −2.1, −0.4; *p* = 0.009. There was a Joinpoint in 2019. This region experienced a more pronounced decline in the latter period (2019–2022) with an APC of −2.6% (95% CI −8.6, 3.8; *p* = 0.354) (Figure 4).

The Caribbean region showed a different pattern, with an overall decrease in the vaccination rate with an APC of −0.7% (95% CI −1.1, −0.4; *p* = 0.001). There was a Joinpoint in 2016 (Figure 5). In the second period (2016–2022), there was a decrease with an APC of −1.1% (95% CI −2.1, −0.1; *p* = 0.031).

In South America, the total period saw a decline in vaccination rates with an APC of −2.5% (95% CI −3.1, −1.8; *p* < 0.001). There was a Joinpoint in 2015 (Figure 6). The trend intensified in the second period (2015–2022), with an APC of −3.1% (95% CI −4.4, −1.8; *p* = 0.001).

These findings suggest region-specific variations within Latin America and the Caribbean region, with South America experiencing the most impactful decline over the study period (Figure 7). The Americas experienced a 4% reduction in DTP3-vaccine coverage from 2019 to 2022. However, vaccine-coverage rates remained unchanged in North America, South America, and the Caribbean. In contrast, Central America witnessed a more substantial decline in vaccine coverage, with a decrease of 7% (Table 3).

Table 4 displays the DTP3 rates for the years 2019 and 2022. On average, there was a reduction of −4.20% across countries, with a standard deviation of 6.08 (*p* < 0.001). In Figure 8, Figure 9 and Figure 10, we present the map of America indicating the absolute differences in vaccine coverage between 2012 and 2022. Between 2019 and 2022, the DTP3-vaccination rates showed a decline, as observed in the dataset, which revealed that twenty-two countries (61.1%) in America registered a decrease, while only six countries (16.7%) indicated an increase, and 8 (22.2%) remained unchanged.

Although COVID-19 impacted vaccine coverage, six countries had increased vaccination coverage after the COVID-19 pandemic. Brazil led this trend with a notable 7% increase in coverage. Following Brazil, Antigua and Barbuda saw a 4% rise, while Jamaica experienced a 2% increase. Canada and Mexico also reported modest gains of 1% each.

Conversely, eight countries demonstrated stability in their DTP3-vaccine coverage during the same period. Countries such as Chile, Costa Rica, Cuba, Haiti, Suriname, Trinidad and Tobago, the United States, and Uruguay maintained their coverage levels, with no percentage change observed. However, the study also identified 22 American countries where there was a decline in DTP3-vaccine coverage (Argentina, Bahamas, Barbados, Belize, Saint Kitts and Nevis, Bolivia, Colombia, Guyana, El Salvador, Nicaragua, Saint Lucia, Paraguay, Saint Vincent and the Grenadines, Ecuador, Honduras, Venezuela, Guatemala, Panama, Grenada, Peru, Dominica, and the Dominican Republic). The Dominican Republic, Guyana, Panama, and Saint Kitts and Nevis experienced a 1% decrease, while Argentina faced a 2% decline. In nations with declining vaccination coverage, the average absolute decrease was −7.37%, with a standard deviation of 5.39 (Table 4) (Figure 8, Figure 9 and Figure 10).

In North America, in two countries, Canada and Mexico, the DTP3-vaccine coverage slightly increased by 1% between 2019 and 2022, while in the United States, it remained the same. In South America, Brazil increased the coverage by 7%, and three countries, Chile, Suriname, and Uruguay, remained at the same level. In the other countries, the coverage decreased. In Central America, except for Costa Rica, whose rate remained equal, all the nations fell in their coverage. The same happened in the Caribbean. All the countries saw decreases; the only exceptions were Antigua, Barbuda, and Jamaica, which increased 4% and 2%, respectively, and Cuba, Haiti, and Trinidad and Tobago, which remained at the same level (Figure 10).

We found that in 2019, a total of 19 countries in the Americas, namely the United States, Canada, Chile, Uruguay, Trinidad and Tobago, Saint Vincent and the Grenadines, Saint Kitts and Nevis, Saint Lucia, Nicaragua, Jamaica, Guyana, Grenada, El Salvador, Dominica, Cuba, Costa Rica, Colombia, Belize, Barbados, and Antigua and Barbuda, had successfully achieved the World Health Organization’s goal of attaining at least 90% coverage for DTP3 vaccine by the year 2030. After the pandemic, by 2022, it was observed that six nations—Barbados, Belize, Colombia, El Salvador, Grenada, and Saint Lucia—no longer met the 90% coverage target for the DTP3 vaccine. This was a change from the list of countries that had achieved this goal in 2019 (Figure 11 and Figure 12).

Twenty countries had a Joinpoint close to 2019 (Table 5, Appendix A). In five countries, Antigua and Barbuda, Costa Rica, Haiti, Peru, and Saint Vincent and the Grenadines, there was a Joinpoint in 2017 (95% IC 2014–2020). In five countries, Canada, Ecuador, Grenada, Jamaica, Nicaragua, and Paraguay, the Joinpoint was in 2018 (95% IC 2016–2019). In three countries, Belize, Colombia, and Saint Lucia, there was a Joinpoint in 2019 (95% IC 2017–2020). Finally, there were six countries, Bahamas, Dominica, Mexico, Saint Kitts and Nevis, Suriname, and Uruguay, with a Joinpoint in 2020 (95% IC 2014–2020). In all these countries, there was a decrease in the APC. The only exceptions were four countries, Canada, Bahamas, Mexico, and Suriname, which managed to achieve an increase in the period after the Joinpoint.

Using interrupted time series analysis, we detected changes in all the regions of America. Table 6 presents the segmented regression analysis of DTP3-vaccine coverage parameters (Figure 13, Figure 14, Figure 15 and Figure 16).

In the Americas (Figure 13), the year’s coefficient was −271,700 (*p* = 0.002), suggesting a decreasing trend in DTP3 vaccination over the years. The COVID-19 coefficient is negative, approximately −3,824,000, indicating a decrease during the pandemic.

In North America, the COVID-19 coefficient is negative, at approximately −960,900 (*p* = 0.140) (Figure 14).

In Latin America and the Caribbean, the coefficient of COVID-19 was −3,408,000 (*p* = 0.078), indicating a decrease during the pandemic (Figure 15).

In Central America, the coefficient of COVID-19 was −73,740 (*p* > 0.05) (Figure 16).

In the Caribbean, the COVID-19 coefficient was negative, −94,750 (*p* = 0.195) (Figure 17).

In the Caribbean, the COVID-19 coefficient was negative, −2,071,000 (*p* > 0.05) (Figure 18). In the segmented regression, we also detected several American countries with fewer children vaccinated with DTP3 after the COVID-19 pandemic. Those countries were Belize, Grenada, Peru, Suriname, the United States, Mexico, Antigua and Barbuda, Argentina, Bahamas, Barbados, Bolivia, Brazil, Canada, Chile, Dominican Republic, Ecuador, El Salvador, Haiti, Honduras, Jamaica, Nicaragua, Panama, Saint Vincent and the Grenadines, Uruguay, and Venezuela (Table 7).

In Belize, the segmented regression model explains a high proportion of the variance in the number of children who received DTP3 doses (Appendix A), with an R^2^ value of 0.937, suggesting a good fit. The COVID-19 variable had a coefficient of a −3833 (*p* < 0.05). This indicates that the COVID-19 pandemic is associated with a decrease in the number of DTP3 vaccinations. The Interaction term has a positive coefficient of roughly 240 (*p* < 0.05). For the data from Grenada (Appendix A), the segmented regression analysis has been conducted, yielding the following results: The R^2^ value of the model was 0.983, indicating that the model explains a remarkably high portion of the variance in the DTP3 vaccination numbers, suggesting a perfect fit. The coefficient for the COVID-19 variable has a value of −837.68 (*p* < 0.05), indicating that the start of the COVID-19 pandemic was associated with a substantial decrease in the number of DTP3 vaccinations. The Year variable has a negative coefficient of −12.09 (*p* = 0.067). This suggests a decline in vaccinations over the years. The Interaction term, which represents the interaction between the year and the occurrence of COVID-19, has a positive coefficient of 41.60 (*p* > 0.05).

The segmented regression model for Peru (Appendix A) has an R^2^ value of 0.803, which is considered a good fit, meaning that the model explains 80.3% of the variance in the DTP3-vaccination numbers. The coefficient for COVID-19 has a value of −372,300 (*p* < 0.05), indicating that the onset of the COVID-19 pandemic was associated with a high decrease in the number of DTP3 vaccinations. The Interaction term between the year and the occurrence of COVID-19 has a coefficient of 34,450 (*p* < 0.05). In the segmented regression analysis for Suriname (Appendix A), the model’s R^2^ value is 0.733, which explains much of the variance in the DTP3-vaccination numbers. The coefficient for COVID-19 is −15,210 (*p* < 0.05), which implies that the onset of the COVID-19 pandemic was associated with a decrease in the number of DTP3 vaccinations. The Interaction term has a coefficient of y 1420 (*p* < 0.05). In Mexico (Appendix A), the segmented regression model has a good fit, with an R^2^ value of 0.884. The coefficient for the Year was −63,130 (*p* < 0.05), indicating a decline in DTP3 vaccinations over the years. The COVID-19 coefficient is negative, with a value of approximately −1,379,000 (*p* = 0.098), suggesting a substantial decrease in vaccinations during the pandemic. The segmented regression analysis for the United States data (Appendix A) provides the following insights: The model has a good fit, with an R^2^ value of 0.848. The coefficient for the Year is −44,760 (*p* < 0.01, suggesting a declining trend in the number of DTP3 vaccinations over the years. The COVID-19 coefficient is negative, with a value of −1,093,000 (*p* = 0.092), which would indicate a substantial decrease in the number of vaccinations during the pandemic. The Interaction has a positive coefficient of 108,000 (*p* = 0.099), indicating a potential interaction effect between the point year and the occurrence of COVID-19. Appendix A summarizes all the main findings to allow for a global view.

## 4. Discussion

Our research focused on examining the effects of the COVID-19 pandemic on DTP-immunization patterns across the Americas. Our findings reveal that vaccination rates have been negatively impacted in various American countries, particularly Central America.

### 4.1. Overview of the Study and Its Context

#### 4.1.1. Methodological Considerations and Data Limitations

Our methodologies included comparing country-specific vaccination coverage between 2019 and 2022 and analyzing trends using joint points and segmented regression during the period of 2011–2022, making our findings more robust.

While acknowledging the constraints of the available data and potential inconsistencies in reporting, the significance of the United Nations Children’s Fund’s dataset for this investigation remains paramount. Despite the United Nations Children’s Fund’s consistent methodology in data collection and reporting, variations in national healthcare infrastructures and reporting systems might lead to discrepancies in data quality and precision, potentially influencing the interpretations of our analysis. Furthermore, the data’s compilation at national and regional scales could mask local nuances in vaccination trends, especially in areas with healthcare-accessibility issues or socioeconomic disparities. Nevertheless, in the last 20 years, global immunization-coverage data quality improved [73]. As of the latest data update, in 2022, our study does not include real-time data from 2023 onwards. Thus, our findings might not fully capture the ongoing impact of the COVID-19 pandemic on DTP-vaccination rates in the Americas. Future research, with access to more current data, would be essential in providing a more comprehensive understanding of these trends.

Among the strengths of our study is that this study employs a robust methodological framework that includes joint points and segmented regression analysis. Using two different methods enhances the reliability and validity of the findings. Leveraging comprehensive data sets provided by the United Nations Children’s Fund, despite the acknowledged limitations, our approach benefits from consistent data collection and reporting methodologies. This is particularly valuable given the varied national healthcare infrastructures and reporting systems across the countries studied. Our analysis offers insightful revelations into regional disparities in vaccination trends, underlining the necessity for customized public health approaches. Such detailed examination allows for a nuanced understanding of vaccine-coverage trends across the Americas, providing a solid foundation for future policy formulation and implementation to mitigate the pandemic’s impact on essential vaccination programs.

Despite the methodological strengths, our study acknowledges certain limitations associated with the available data, potentially influencing our analysis’s robustness. The variance in data quality and precision due to differences in national healthcare infrastructures and reporting systems poses a challenge, necessitating a cautious approach to interpreting our findings. A broader comparative analysis incorporating other regions, such as Africa and Europe, could have enriched our understanding of global patterns and identified unique challenges and successful strategies in different contexts. Furthermore, the impact of the digital divide on data reporting accuracy and timeliness, especially in regions with limited access to technological resources, remains a critical area that warrants deeper exploration. Addressing these weaknesses would not only enhance the credibility of our study but also provide more comprehensive insights, supporting the development of more effective vaccination strategies and public health policies.

In addressing the methodological considerations of our study, it is important to highlight our dual-analytical approach, employing both Joinpoint regression and segmented regression analyses. This decision was underpinned by the objective to enhance the robustness of our findings, allowing for the identification of trend changes in DTP3-vaccination coverage without prior assumptions (Joinpoint regression) and the assessment of the direct impact of the COVID-19 pandemic with predefined intervention points (segmented regression). The complementary nature of these methods strengthens our analysis, as corroborated by the literature suggesting the value of utilizing multiple statistical approaches in public health research to ensure the validity of results [74]. However, it is crucial to acknowledge potential data limitations inherent in our study, such as the reliability of reported vaccination rates and the assumption that detected trend changes are solely attributable to the pandemic, without discounting other concurrent public health interventions or socioeconomic factors. These considerations underscore the complexity of interpreting trend data in the context of global health crises and the necessity of a cautious approach in attributing causality.

A possible methodological concern could be the statistical power of using Joinpoint regression analysis with only 11 data points. It is essential to highlight that, contrary to the assumption of insufficient data for robust analysis, this number of data points is quite suitable for detecting a single joinpoint. According to the statistical guidelines provided by the National Institutes of Health, for datasets comprising 7 to 11 data points, the default maximum number of joinpoints recommended is one [75]. To detect two Joinpoints, the number of data points should be between 12 and 16. Therefore, in the context of our study with 11 data points, applying a single joinpoint analysis falls well within these recommended parameters, providing a statistically sound approach for identifying trend changes in the data. This approach is further justified because our study aims to detect a single critical shift in the trend rather than multiple joinpoints, making our dataset adequate and optimal for the intended analysis.

#### 4.1.2. Discrepancies in Vaccine-Coverage Data and Trend Analysis

There appears to be a disparity between the number of countries in the Americas that experienced a drop in immunization rates between 2019 and 2022, totaling 22, and the smaller subset of 16 countries within this group where a decrease in vaccination trends was observed. This variation can be attributed to the fact that, despite the reduction in vaccine coverage in particular countries over the observed years, the decrease is not yet significant enough to establish a trend shift in some of these nations due to the small number of births in some countries. Additionally, the discrepancy in results between Joinpoint and segmented regression may also be influenced by the methodologies employed: Joinpoint regression identifies year-significant trend changes but with a 95% confidence interval that spans a range of years, whereas segmented regression focuses on a precise time point. This difference in approach to determining trend changes at specific times versus a range of years can lead to varying interpretations in the vaccination-coverage data [76,77,78].

### 4.2. Analysis of DTP3-Vaccination Trends

The data analysis shows a declining trend in vaccination coverage throughout the Americas, possibly due to socioeconomic and cultural factors and vaccine hesitancy, which COVID-19 exacerbated [79,80].

In the context of DTP3-vaccine coverage between 2019 and 2022, several American countries exhibited an increase in vaccine coverage. Despite the challenging times of the COVID-19 pandemic, these enhancements in vaccine coverage highlight the effectiveness of public health interventions and the resilience of healthcare systems in these countries. Conversely, several countries demonstrated stability in their DTP3-vaccine coverage during the same period. Amidst the global health crisis, the stability in coverage underscores the strength and consistency of vaccination programs in these nations. It suggests that these countries successfully navigated the complexities introduced by the pandemic, ensuring uninterrupted vaccine delivery to their populations.

However, the study also identified countries with declining DTP3-vaccine coverage. These reductions highlight the challenges and disruptions caused by the COVID-19 pandemic, potentially reflecting resource reallocation, access issues, or public hesitancy toward vaccination. It underscores the need for focused efforts to strengthen and adapt vaccination programs in the face of such unprecedented global health challenges.

Our analysis revealed that DTP3 rates remained constant throughout the 2019–2022 pandemic in most North American nations, as shown in Table 4. In this period, Canada and Mexico exhibited a minor increase in DTP3 rates, whereas the United States maintained its existing level of coverage [4]. Some countries did not have their coverage affected at the end of the pandemic. These results could indicate the efficiency of the national vaccination programs and the resilience of healthcare systems in these countries, enabling them to sustain their immunization rates in the face of numerous challenges presented by the persistent COVID-19 pandemic globally.

Nevertheless, there are differences in subnational units. For example, In Haiti, a significant decrease in DPT3 service volume, amounting to 5% or more, was observed in 80% of the subnational regions during the third quarter of 2022 [81].

The reduction in coverage in some American nations might be ascribed to enduring issues in the country’s health system infrastructure, particularly shortcomings in the vaccine-distribution network, a situation potentially aggravated by the extra burden imposed by the COVID-19 pandemic [82,83,84,85,86,87].

Our study has revealed notable reductions in immunization rates across various American nations. Additionally, we observed a shift in the trend of vaccine coverage in Latin America and the Caribbean. This trend shift, marked by an accelerated decline in coverage, was evident in all subregions. Our findings are consistent with the literature indicating that DTP3 rates dropped by 5.06% in the Caribbean and Latin America between 2019 and 2021 [35].

### 4.3. Comparative Global and Regional Perspectives

The global decline in vaccine coverage, spurred by the COVID-19 pandemic, extends beyond DTP vaccines. From January to December 2020, approximately 30 million children missed DTP3 vaccinations, and 27.2 million missed MCV1 vaccinations [32]. It has been reported that there was also a reduction in HPV in some parts of the United States [79]. The worldwide coverage of DTP3 experienced a decline of 5.81% from 2019 to 2021, decreasing from 86% to 81% [51]. In considering the broader implications of our findings, it is crucial to assess the impact of the digital divide on data reporting. In regions of the Americas with limited access to technological resources, the accuracy and timeliness of data reporting may be compromised. This factor is crucial in understanding the regional disparities in vaccine coverage and the challenges in data collection.

The decrease in DTP coverage is a global problem. The global coverage of DTP3 decreased from 86% in 2019 to 83% in 2020 [51,88]. Similar declines had been found in Africa and Asia [44,89]. In contrast to other regions, Europe experienced a minor reduction in vaccination coverage. Specifically, the coverage for the third dose of the diphtheria–tetanus–pertussis (DTP3) vaccine saw a marginal decline of 1.05% from 2019 to 2021. This resulted in a slight dip in coverage rates, from 95% in 2019 to 94% in 2020. Furthermore, a comparative analysis with other regions, such as Africa or Europe, could offer valuable insights into global patterns and region-specific challenges in maintaining vaccination during the pandemic. Such a comparative perspective could help identify unique challenges and successful strategies in different regions, offering lessons for future healthcare planning and policy formulation.

### 4.4. Influential Factors and Challenges

Socioeconomic factors have been pivotal in influencing vaccine accessibility during the pandemic. Economic challenges exacerbated by the pandemic have widened existing disparities in healthcare access, further impacting vaccination rates. A detailed examination of these socioeconomic variables provides a more nuanced understanding of the vaccination landscape across different socio-demographic groups.

Numerous elements could account for the shift in vaccination coverage. Factors at the national level, like elevated fertility rates, combined with community-specific factors, such as widespread illiteracy, play a role in the higher incidence of unvaccinated children [33,90]. Research conducted in the US revealed that for all types of hepatitis vaccines, both adherence and completion rates are notably low, exhibiting considerable differences across various socio-demographic and clinical profiles. The likelihood of poor adherence and incomplete vaccination was typically linked to factors such as being male, belonging to a younger age group, identifying as Black or Hispanic, and having lower levels of education and household income [91].

In Africa, incomplete vaccinations are primarily due to caregivers’ time constraints, limited immunization knowledge, vaccine or staff shortages at health facilities, missed vaccination chances, concerns about side effects, poor access to services, and caregiver vaccination beliefs [92]. A recent study shows that 35.5% of African populations have incomplete immunization, with home births, rural residency, a lack of prenatal care, limited immunization awareness, and maternal illiteracy being the main risk factors [93].

Additionally, American countries’ varied public health policies and COVID-19 response strategies have potentially influenced DTP-vaccination rates [50]. Redirecting healthcare resources from routine vaccinations to COVID-19-response efforts in some regions may have contributed to the observed decline in vaccine coverage.

These reductions emphasize the significant influence of the COVID-19 pandemic on regular vaccination programs within the Americas, signaling a need for focused efforts to recover and enhance vaccine coverage. COVID-19 had an impact on children’s vaccination. The decline in vaccination rates can be linked to various factors associated with the COVID-19 pandemic, including the reallocation of healthcare resources to address the outbreak, the implementation of lockdowns, and the imposition of restrictions on movement that impeded access to vaccination services [94,95,96,97]. In a survey among parents of children aged 0–4 years across America, Europe, and Australia, 83% of respondents considered it crucial for their child to receive the recommended vaccines despite the COVID-19 pandemic. About half of the routine vaccine appointments were postponed or canceled due to the pandemic. However, 61% of parents indicated their desire to make up for missed vaccinations once COVID-19 restrictions were eased or lifted [98].

### 4.5. Strategies and Interventions for Enhancing Vaccine Coverage

Interestingly, despite the pandemic’s difficulties, some nations, including Canada, Mexico, Jamaica, Antigua and Barbuda, and Brazil, were able to increase their vaccination coverage. It is essential to highlight the situation in Brazil, where the DPT3 rate increased by 7% towards the conclusion of the COVID-19 pandemic. This rise could be credited to successful immunization strategies, focused interventions, or heightened governmental backing for vaccination initiatives during the crisis. Analyzing and drawing lessons from the experiences of these nations is essential for enhancing immunization services in future crises.

As there are many incomplete vaccinations and parents are willing to catch up, healthcare services should try to reach parents [4]. Strategies need to be developed to increase DTP3-vaccination coverage. One strategy that has been suggested is co-administration with other vaccines [99].

The disparities observed across different countries emphasize the importance of developing specific strategies to boost vaccination rates, particularly in nations experiencing a downturn in these trends. The persistent oversight of vaccination initiatives, addressing challenges within healthcare systems, and active community involvement are crucial for sustaining and improving immunization rates in these areas.

Our findings are consistent with previous studies showing that the worldwide COVID-19 pandemic has interrupted crucial health services, including vaccination efforts, and has been linked to growing inequities [100,101,102]. Moreover, concerns about contracting the virus and vaccine hesitancy might have contributed to parents’ hesitancy in seeking healthcare facilities for their children’s vaccinations [103,104]. Healthcare workers, especially community nurses, play an essential role [20].

Last, policies should enhance public awareness about the importance of routine vaccinations and combating misinformation. Community-based interventions and social media campaigns can play a crucial role in increasing vaccine uptake [14,105].

### 4.6. The Role of Healthcare Systems and Planning

The disruption caused by the COVID-19 pandemic has underscored the importance of having robust health systems capable of maintaining vital medical services during challenging times [106]. Research has shown statistically significant differences in monthly vaccination rates between rural and urban regions [107]. The differences in vaccination trends across regions emphasize the necessity for customized public health approaches, called precision public health [107,108]. Policies must be region-specific, considering each area’s unique challenges and resources.

Initiatives must be undertaken to fortify immunization programs and enhance vaccine coverage throughout the continent, especially in the areas most impacted by the decline in vaccination rates [109]. The decrease in vaccination rates, especially in areas with weaker healthcare infrastructures, underscores the need to strengthen health systems, ensure adequate funding, train healthcare workers, and improve vaccine supply chains.

### 4.7. Looking to the Future

While our study conclusively demonstrates the impact of the COVID-19 pandemic on vaccine coverage across America, it is important to acknowledge that it did not explicitly evaluate the effects of geographic location, socioeconomic factors, or the consistency of maternal prenatal care on these trends. Future research should aim to dissect these variables to provide a more nuanced understanding of their contributions to vaccine-coverage disparities observed during the pandemic period.

The decline in vaccination coverage has profound implications, such as the potential for outbreaks of diseases like diphtheria, tetanus, and pertussis, further straining healthcare systems amid the COVID-19 pandemic. Addressing the fall in immunization rates is critical, as the risks of postponing vaccinations are more significant than those associated with COVID-19 during routine immunizations [110].

Efforts to enhance immunization rates must consider factors like healthcare access, vaccine supply issues, and government commitment. The decrease in countries meeting WHO’s DTP targets, particularly in Latin America and the Caribbean, underscores the need for targeted interventions. Additionally, even with improved vaccination programs, the time required for catch-up immunizations could be extended, raising the risk of disease outbreaks [111,112]. Prompt and efficient vaccine distribution and the recovery of coverage levels are essential. Collaboration among governments, healthcare bodies, and international agencies is crucial to preserve vaccination progress and protect populations from vaccine-preventable diseases [113,114,115,116]. We should plan for the next pandemic, analyzing and replicating the strategies of countries that have successfully increased or maintained high vaccination rates post-pandemic, which could be beneficial. It is also essential to strengthen epidemiology services related to preventing vaccine-preventable diseases. The continuous monitoring and evaluation of vaccination programs are crucial. Identifying and addressing issues promptly can avoid further declines in vaccination rates.

Despite initial fears and logistical challenges like transportation restrictions, community commitment to vaccination during the pandemic remained strong in many countries. Systemic changes within healthcare, such as staff reallocations, caused temporary delays but were efficiently addressed. Notably, some community misconceptions about the pandemic existed but did not significantly deter vaccination efforts, reflecting a deep-rooted trust in the importance of vaccines [117]. It is crucial to monitor subnational units to avoid inequities [81].

## 5. Conclusions

This study comprehensively assessed the impact of the COVID-19 pandemic on DTP3 vaccination coverage in the Americas from 2012 to 2022. The findings indicate a significant and concerning decline in vaccination rates, particularly after 2019. The overall APC for the entire period was −1.4%, with an absolute decrease in vaccine coverage of −4% from 2019 to 2022. Notably, the most substantial relative decrease occurred in Central America, with a 7.87% reduction in coverage.

Interestingly, while the pandemic adversely affected vaccination coverage in most countries, some nations showed increased DTP3 vaccine coverage post-pandemic, and several countries maintained stable vaccination rates. The study’s results are alarming, considering the importance of maintaining high DTP-vaccination coverage to prevent outbreaks of vaccine-preventable diseases. The pandemic’s disruptive impact on public health initiatives is underscored by the decline in coverage rates, particularly in countries that had previously met the WHO target of 90% coverage. This decline in vaccination rates highlights the need for renewed efforts in vaccination campaigns and public health strategies to reverse the negative trends and ensure the well-being of populations across the Americas.

## Figures and Tables

**Figure 1 vaccines-12-00238-f001:**
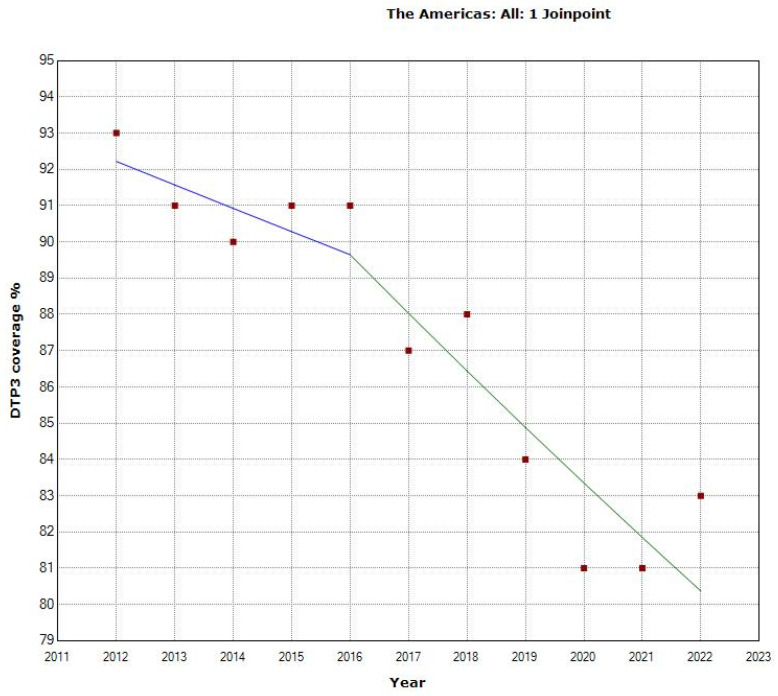
Joinpoint graph of DTP3 in the Americas, 2012–2022 indicate joinpoints at the transitions between colored lines. (Statistical Power: 0.999).

**Figure 2 vaccines-12-00238-f002:**
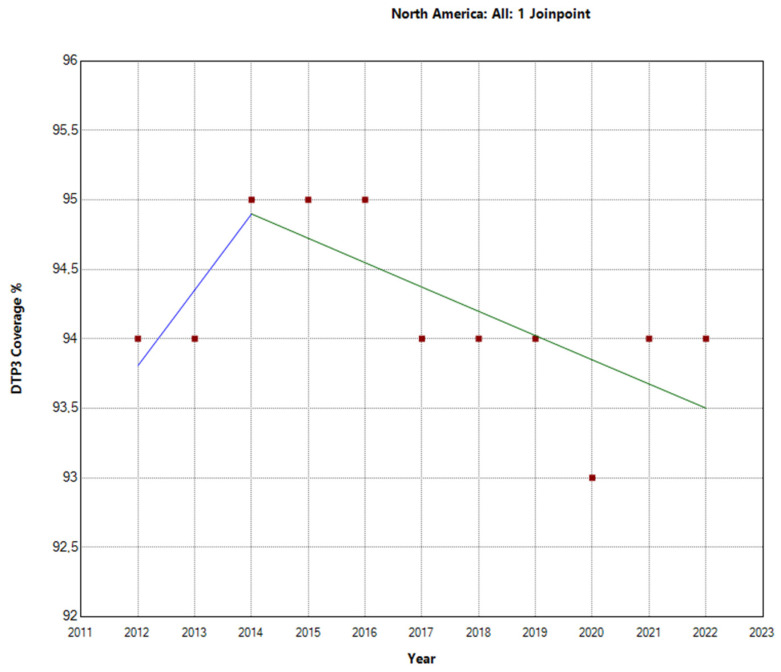
Joinpoint graph of DTP3 in North America, 2012–2022 indicate joinpoints at the transitions between colored lines. (Power = 0.0303).

**Figure 3 vaccines-12-00238-f003:**
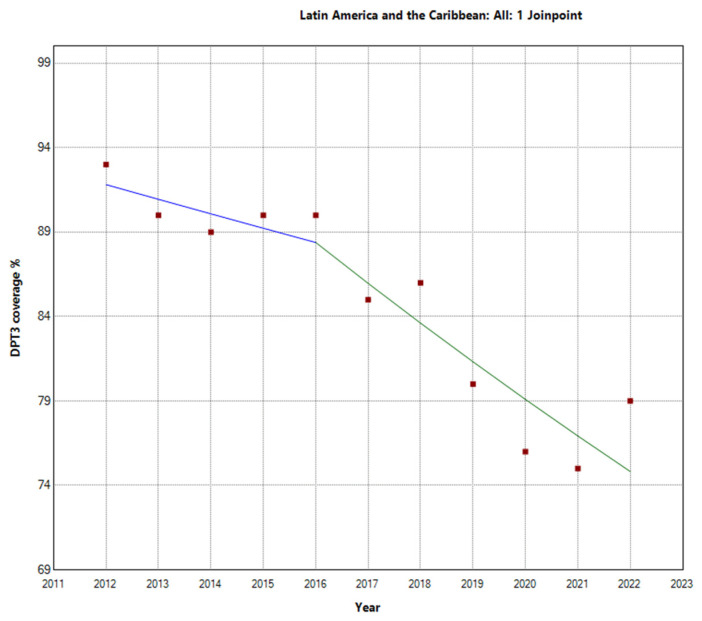
Joinpoint graph of DTP3 in Latin America and the Caribbean, 2012–2022 indicate joinpoints at the transitions between colored lines. (Power = 0.9999).

**Figure 4 vaccines-12-00238-f004:**
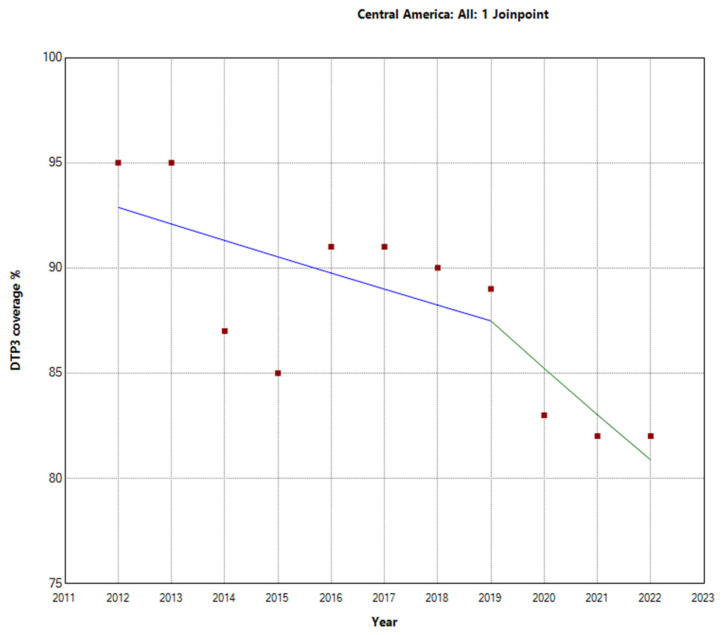
Joinpoint analysis of DTP3-vaccination trends in Central America, 2012–2022 indicate joinpoints at the transitions between colored lines. (Statistical Power: 0.954).

**Figure 5 vaccines-12-00238-f005:**
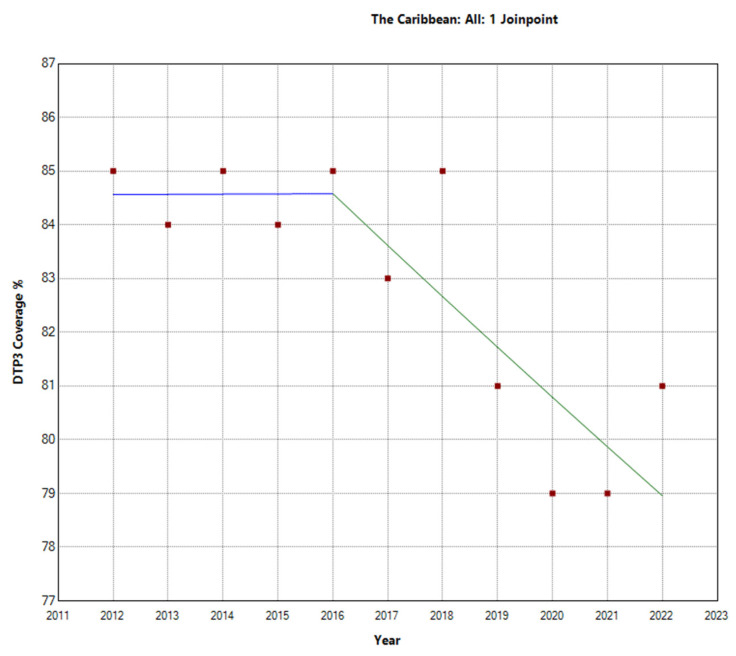
Joinpoint graph of DTP3 in the Caribbean, 2012–2022 indicate joinpoints at the transitions between colored lines. (Statistical Power: 0.977).

**Figure 6 vaccines-12-00238-f006:**
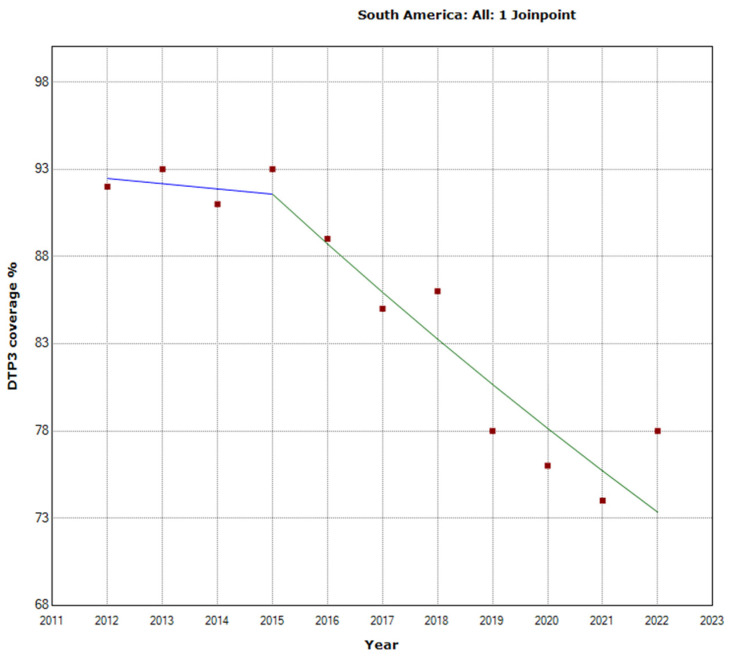
Joinpoint graph of DTP3 in South America 2012–2022 indicate joinpoints at the transitions between colored lines. (Statistical Power: 0.999).

**Figure 7 vaccines-12-00238-f007:**
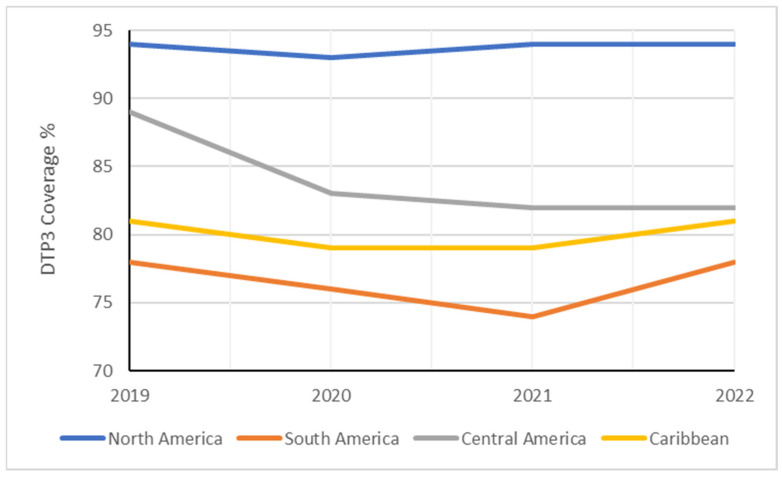
Evolution of DTP3 rates (2019–2022) in South America by region.

**Figure 8 vaccines-12-00238-f008:**
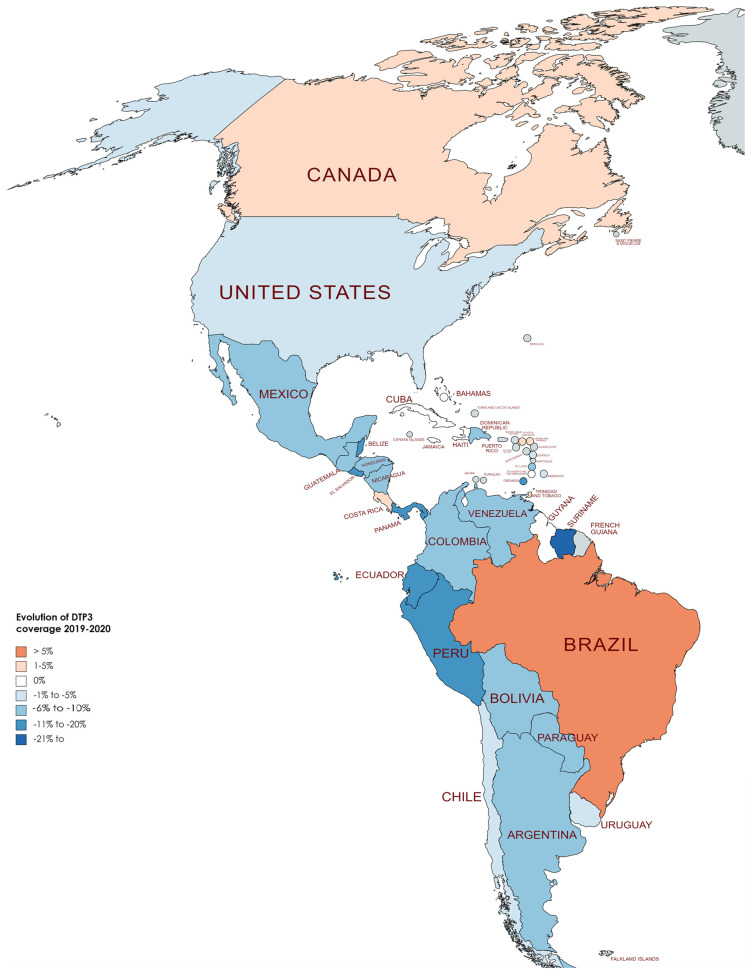
Changes in DTP3 coverage (%) in America, 2019–2020.

**Figure 9 vaccines-12-00238-f009:**
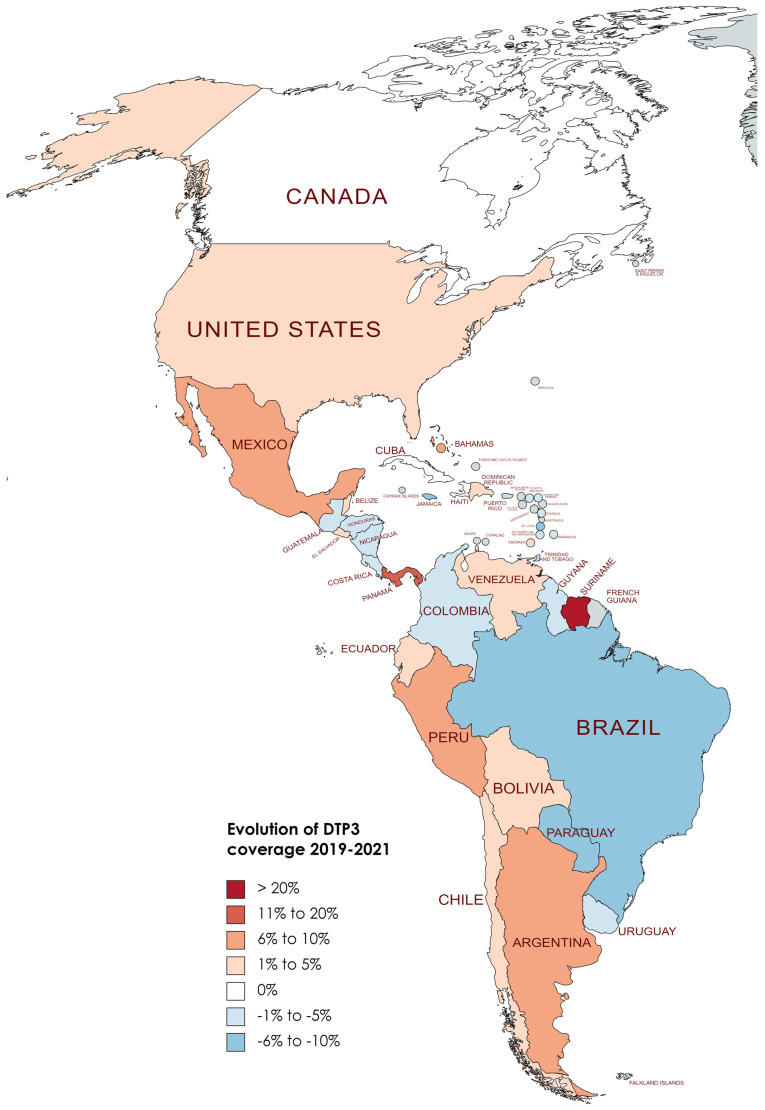
Changes in DTP3 coverage (%) in America, 2019–2021 (Gray meaning no data or excluded).

**Figure 10 vaccines-12-00238-f010:**
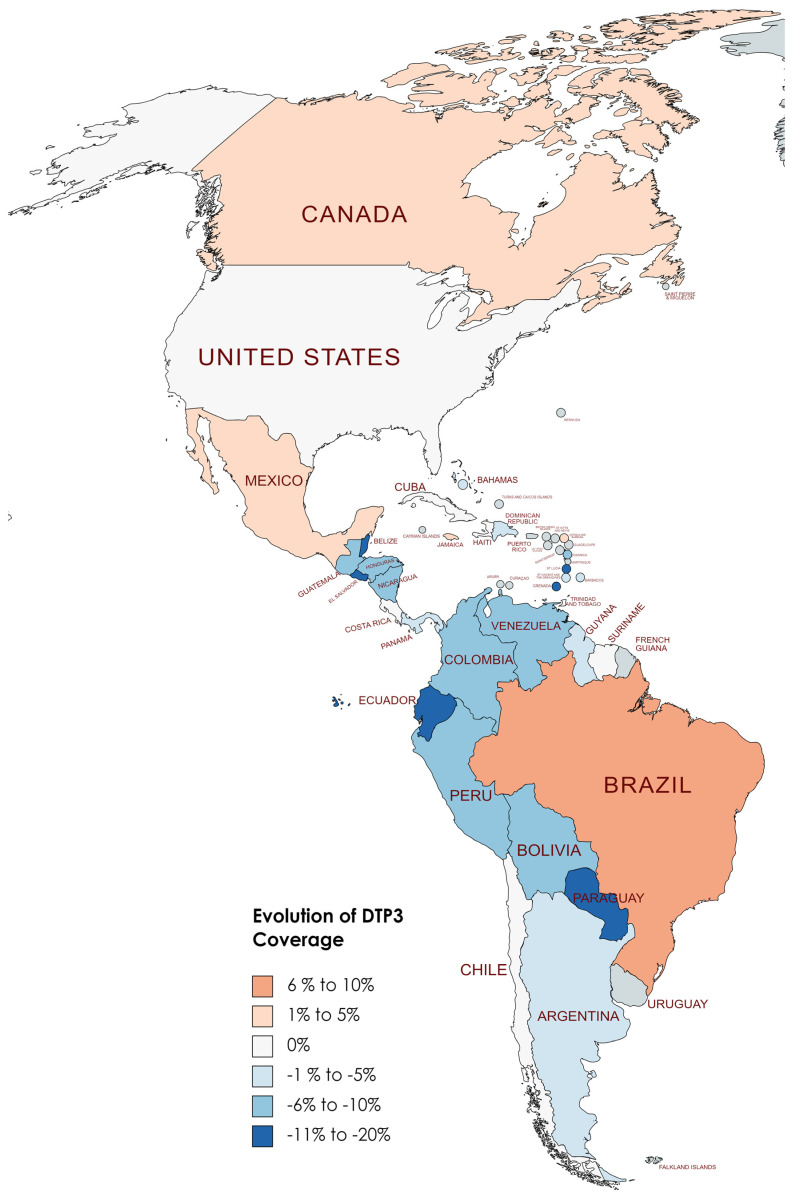
Changes in DTP3 coverage (%) in America, 2019–2022.

**Figure 11 vaccines-12-00238-f011:**
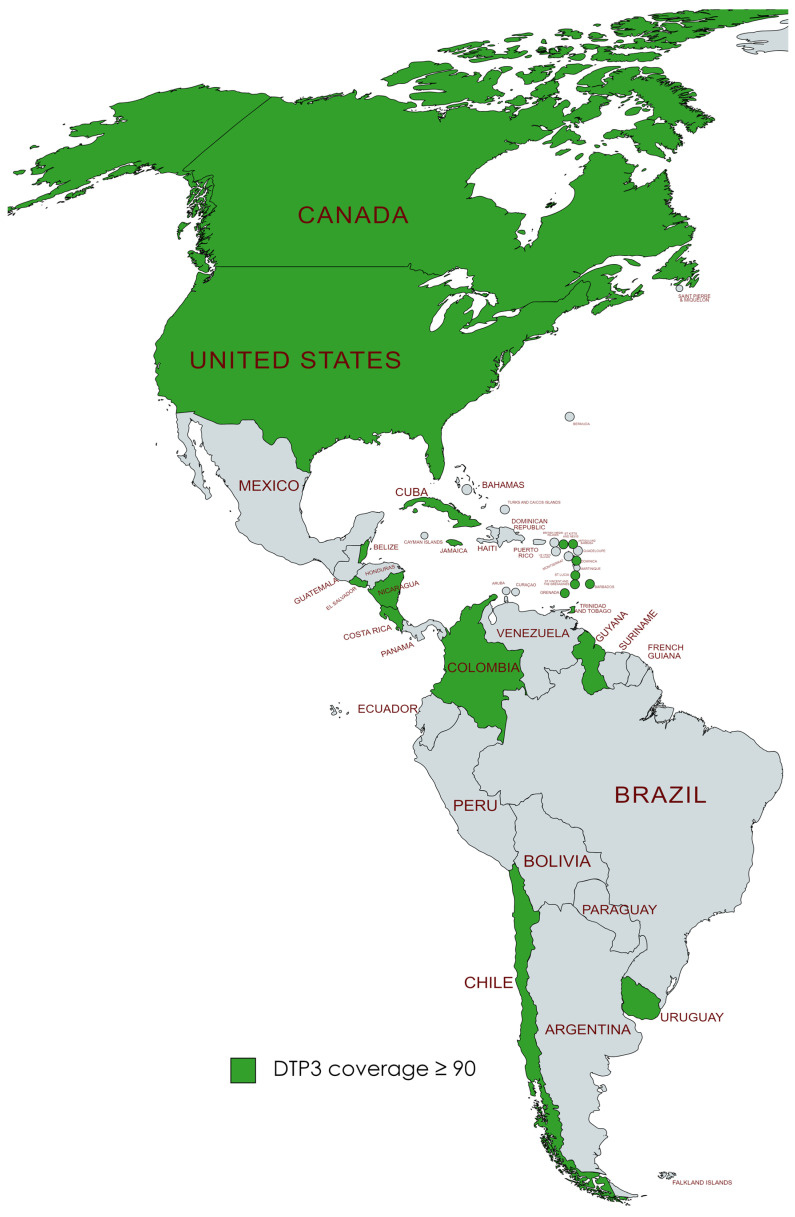
American countries meeting the WHO 90% DTP3-vaccination-coverage target in 2019. (Gray color meaning not reaching the target).

**Figure 12 vaccines-12-00238-f012:**
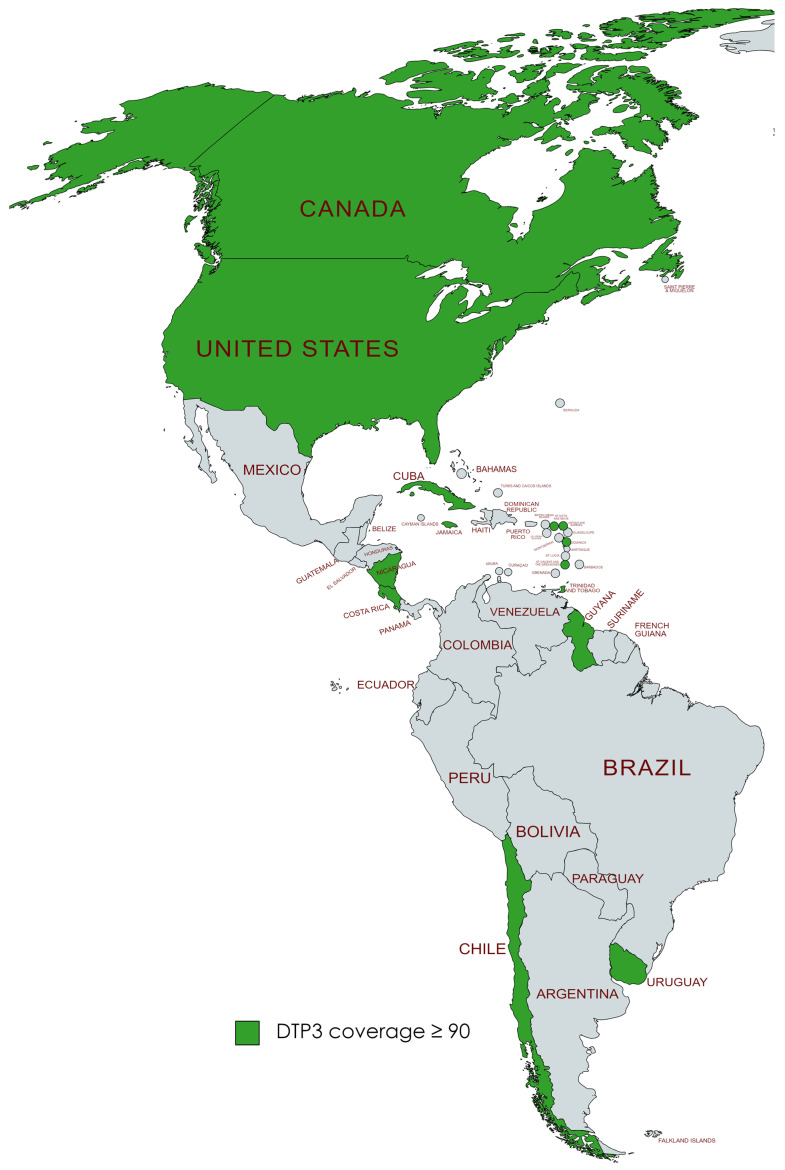
American countries meeting the WHO 90% DTP3-vaccination-coverage target in 2022. (Gray color meaning not reaching the target).

**Figure 13 vaccines-12-00238-f013:**
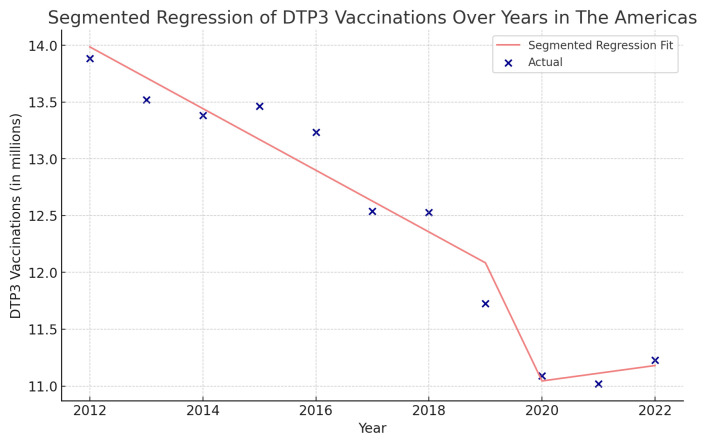
Segmented regression analysis of DTP3 in the Americas, representing the actual DTP3-vaccination numbers (in dark blue) against the predicted values from the segmented regression (in light coral).

**Figure 14 vaccines-12-00238-f014:**
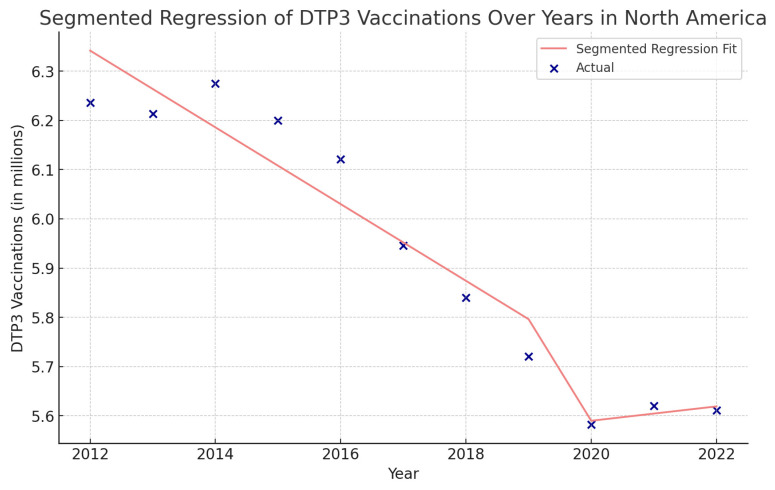
Segmented regression analysis of DTP3 in North America, representing the actual DTP3-vaccination numbers (in dark blue) against the predicted values from the segmented regression (in light coral).

**Figure 15 vaccines-12-00238-f015:**
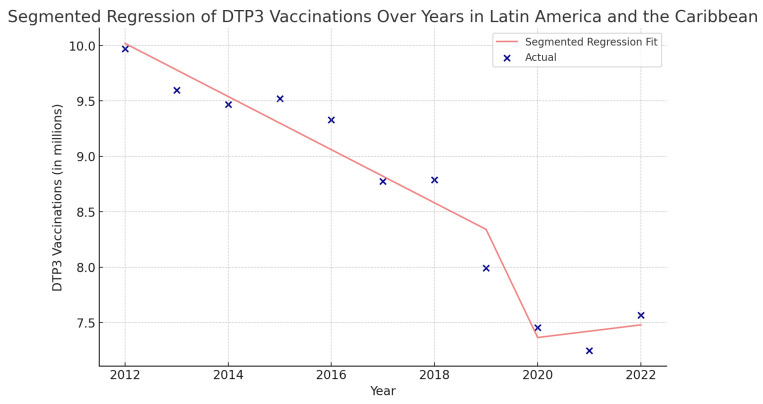
Segmented regression analysis of DTP3 vaccination in Latin America and the Caribbean. The actual DTP3-vaccination numbers (in dark blue) against the predicted values from the segmented regression (in light coral).

**Figure 16 vaccines-12-00238-f016:**
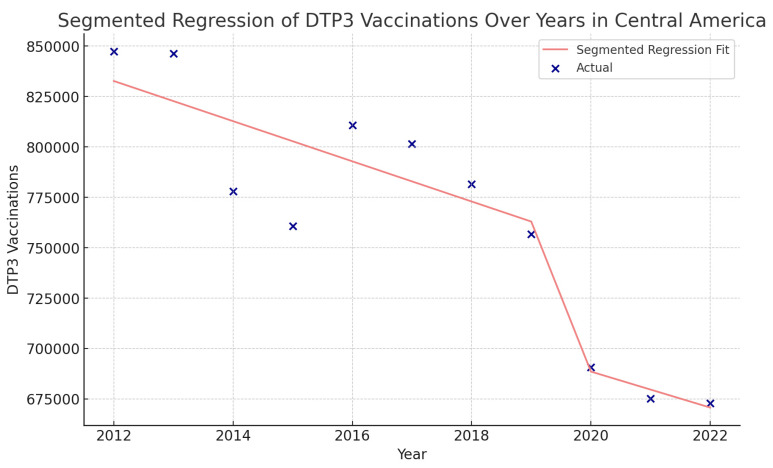
Segmented regression analysis of DTP3 vaccination in Central America. The actual DTP3 (in dark blue) against the predicted values from the segmented regression (in light coral).

**Figure 17 vaccines-12-00238-f017:**
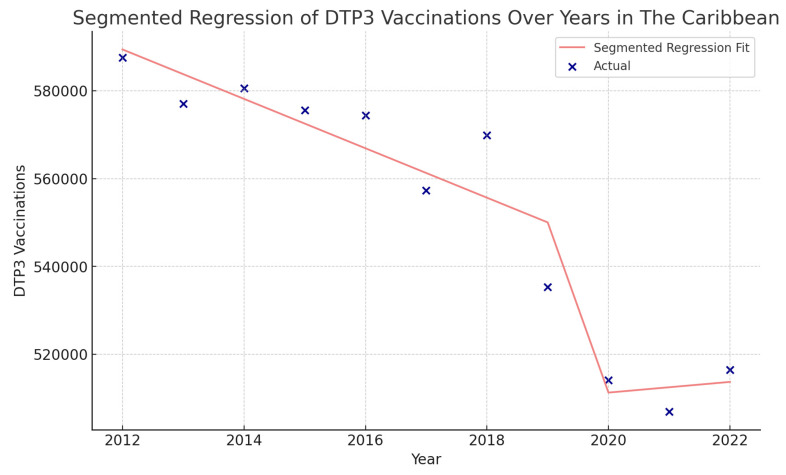
Segmented regression analysis of DTP3 vaccination in the Caribbean. The actual DTP3 (in dark blue) against the predicted values from the segmented regression (in light coral).

**Figure 18 vaccines-12-00238-f018:**
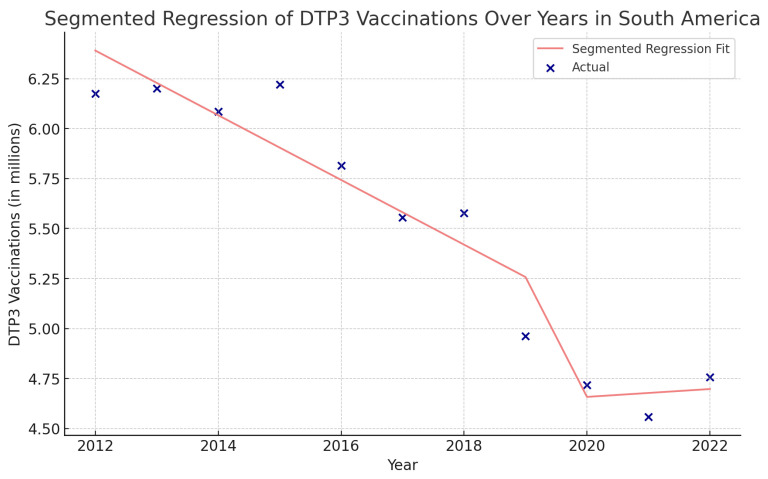
Segmented regression analysis of DTP3 vaccination in South America. The actual DTP3 numbers (in dark blue) are represented against the predicted values from the segmented regression (in light coral).

**Table 1 vaccines-12-00238-t001:** Joinpoint analysis for the third doses of DTP-vaccination rates in the Americas, 2012–2022.

Periods	Years	APC (95% CI)	*p*
Total Period	2012–2021	−1.4 (−1.8; −1.0)	<0.001
Period 1	2012–2016	−0.7 (−2.9; 1.5)	0.464
Period 2	2016–2022	−1.8 (−2.9; −0.7)	0.008

**Table 2 vaccines-12-00238-t002:** DTP3 Joinpoint in American regions, 2012–2022.

Periods	Years	APC	95% LCI	95% UCI	*p*
North America					
Total Period	2012–2022	−0.1	−0.2	0	0.136
Period 1	2012–2014	0.6	−0.2	1.4	0.494
Period 2	2014–2022	−0.2	−0.9	0.1	0.057
Latin America and the Caribbean					
Total Period	2012–2022	−2.1	−2.7	−1.5	<0.001
Period 1	2012–2016	−0.9	−4.4	2.6	0.531
Period 2	2016–2022	−2.7	−4.5	−0.9	0.010
Central America					
Total Period	2012–2022	−1.3	−2.1	−0.4	0.009
Period 1	2012–2019	−0.9	−2.8	1.1	0.321
Period 2	2019–2022	−2.6	−8.6	3.8	0.354
Caribbean					
Total Period	2012–2022	−0.7	−1.1	−0.4	0.001
Period 1	2012–2016	0.0	−1.9	2.0	0.996
Period 2	2016–2022	−1.1	−2.1	−0.1	0.031
South America					
Total Period	2012–2022	−2.5	−3.1	−1.8	<0.001
Period 1	2012–2015	−0.3	−5.9	5.6	0.895
Period 2	2015–2022	−3.1	−4.4	−1.8	0.001

**Table 3 vaccines-12-00238-t003:** Changes in DTP3 rate (%) in American regions between 2019 and 2022.

Region	2019	2020	2021	2022	Absolute Changes 2019–2022	Relative Changes 2019–2022
Americas	84	81	81	83	−1%	−1.19%
North America	94	93	94	94	0%	0.00%
Latin America and the Caribbean	80	76	75	79	−1%	−1.25%
South America	78	76	74	78	0%	0.00%
Central America	89	83	82	82	−7%	−7.87%
Caribbean	81	79	79	81	0%	0.00%

**Table 4 vaccines-12-00238-t004:** Changes in DTP3 rates (%) across American countries in 2019 and 2022.

	2019		2022		
Country	DTP3 (%)	Births (n)	DTP3 (%)	Births (n)	*p* ^†^
Antigua and Barbuda	95	1058	99	1124	<0.001
Argentina	83	661,385	81	627,741	<0.001
Bahamas	89	4641	87	4659	ns
Barbados	90	3050	86	3037	ns
Belize	98	7333	84	7193	0.003
Bolivia	75	263,006	69	264,070	<0.001
Brazil	70	2,886,359	77	2,723,266	<0.001
Canada	91	363,393	92	376,188	ns
Chile	96	224,350	96	230,824	ns
Colombia	94	733,940	87	723,264	<0.001
Costa Rica	95	65,282	95	60,517	ns
Cuba	99	110,404	99	99,693	ns
Dominica	99	946	92	966	ns
Dominican Republic	89	210,196	88	203,625	ns
Ecuador	85	300,075	70	298,666	<0.001
El Salvador	90	103,547	75	100,313	<0.001
Grenada	94	2041	77	1960	0.054
Guatemala	85	391,582	79	372,335	<0.001
Guyana	99	16,898	98	16,129	ns
Haiti	51	271,669	51	268,523	ns
Honduras	88	215,469	78	217,590	<0.001
Jamaica	96	33,815	98	32,663	ns
Mexico	82	1,965,139	83	1,866,399	<0.001
Nicaragua	98	142,181	92	139,164	<0.001
Panama	88	77,044	87	76,637	ns
Paraguay	86	139,138	69	137,960	<0.001
Peru	88	591,025	82	592,156	<0.001
Saint Kitts and Nevis	97	607	96	563	ns
Saint Lucia	92	2113	81	2035	ns
Saint Vincent and the Grenadines	97	1405	92	1324	ns
Suriname	77	11,049	77	11,123	ns
Trinidad and Tobago	93	18,904	93	17,429	ns
Uruguay	94	3,756,830	94	3,726,867	ns
United States	94	37,107	94	35,668	ns
Venezuela	64	497,361	56	438,384	<0.001

ns = not significant, ^†^ Chi-square Test.

**Table 5 vaccines-12-00238-t005:** Joinpoint analysis DTP3 rates in America, 2012–2022.

Country	Years	APC	95% LCI	95% UCI	*p*
Antigua and Barbuda					
Total Period	2012–2022	−0.4	−0.9	0	0.04
Period 1	2012–2017	−1.2	−2.7	0.4	0.118
Period 2	2017–2022	0.3	−1.3	1.9	0.677
Bahamas					
Total Period	2012–2022	−1.5	−2	−0.9	<0.001
Period 1	2012–2020	−1.8	−2.2	−1.4	<0.001
Period 2	2020–2022	0.6	−2.7	3.9	0.697
Belize					
Total Period	2012–2022	−1.6	−2.6	−0.6	0.005
Period 1	2012–2019	−0.8	−1.7	0.2	0.09
Period 2	2019–2022	−4.4	−8.3	−0.3	0.04
Canada					
Total Period	2012–2022	0.1	0.1	0.2	0.003
Period 1	2012–2018	0	−0.1	0.1	0.931
Period 2	2018–2022	0.3	0.1	0.6	0.008
Colombia					
Total Period	2012–2022	−0.4	−0.9	0.1	0.102
Period 1	2012–2019	0.3	0	0.6	0.071
Period 2	2019–2022	−2.6	−3.8	−1.4	0.002
Costa Rica					
Total Period	2012–2022	−0.4	−0.9	0.1	0.102
Period 1	2012–2017	0.8	−0.6	2.3	0.198
Period 2	2017–2022	−0.2	−1.6	1.2	0.702
Dominica					
Total Period	2012–2022	−0.4	−1	0.1	0.101
Period 1	2012–2020	−0.2	−1.3	0.9	0.677
Period 2	2020–2022	−2.2	−12.5	9.3	0.635
Ecuador					
Total Period	2012–2022	−1.9	−3.2	−0.7	0.007
Period 1	2012–2018	−0.3	−3.2	2.8	0.843
Period 2	2018–2022	−4.8	−10.1	0.8	0.078
Grenada					
Total Period	2012–2022	−2.7	−4.1	−1.3	0.002
Period 1	2012–2018	−0.5	−3.5	2.6	0.71
Period 2	2018–2022	−6.5	−12.1	−0.6	0.036
Haiti					
Total Period	2012–2022	−3.0	−4.3	−1.7	0.001
Period 1	2012–2017	−0.8	−5.2	3.7	0.663
Period 2	2017–2022	−5.1	−9.3	−0.7	0.029
Jamaica					
Total Period	2012–2022	0.2	−0.3	0.7	0.476
Period 1	2012–2018	0.4	−0.8	1.6	0.427
Period 2	2018–2022	−0.7	−6	4.8	0.756
Mexico					
Total Period	2012–2022	−1.7	−3.1	−0.2	0.027
Period 1	2012–2020	−2.3	−4.5	0	0.048
Period 2	2020–2022	2.4	−16.3	25.2	0.783
Nicaragua					
Total Period	2012–2022	−0.9	−1.5	−0.4	0.006
Period 1	2012–2018	0.1	−1	1.2	0.875
Period 2	2018–2022	−2.7	−4.9	−0.6	0.022
Paraguay					
Total Period	2012–2022	−2.8	−4.1	−1.5	0.001
Period 1	2012–2018	−0.3	−1.1	0.6	0.506
Period 2	2018–2022	−7.2	−8.8	−5.6	<0.001
Peru					
Total Period	2012–2022	−1.5	−2.6	−0.4	0.012
Period 1	2012–2017	−0.8	−5.4	4.1	0.709
Period 2	2017–2022	−2.3	−6.8	2.5	0.284
Saint Kitts and Nevis					
Total Period	2012–2022	0	−0.3	0.2	0.87
Period 1	2012–2020	0.1	−0.3	0.6	0.489
Period 2	2020–2022	−1.2	−7.1	5.2	0.661
Saint Lucia					
Total Period	2012–2022	−2.2	−3.2	−1.1	0.001
Period 1	2012–2019	−1.7	−4.1	0.8	0.141
Period 2	2019–2022	−3.8	−13.2	6.7	0.4
Saint Vincent and the Grenadines					
Total Period	2012–2022	−0.5	−0.9	0	0.045
Period 1	2012–2017	0.6	−0.2	1.5	0.105
Period 2	2017–2022	−1.6	−2.4	−0.8	0.003
Suriname					
Total Period	2012–2022	−1	−3.3	1.4	0.361
Period 1	2012–2020	−2.1	−4	−0.2	0.038
Period 2	2020–2022	7.3	−14	33.9	0.466
Uruguay	2012–2022	−0.3	−0.6	−0.1	0.016
Period 1	2012–2020	−0.4	−0.9	0	0.049
Period 2	2020–2022	0.5	−4.5	5.8	0.817

**Table 6 vaccines-12-00238-t006:** Segmented regression analysis of regional DTP3-vaccine coverage.

Region	Intercept	Year	COVID-19	Interaction
The Americas	14,257,167	−271,700 ***	−3,824,000 ^†^	339,600 ^†^
North America	6,419,470	−77,900 ***	−960,900	92,490
Latin America and the Caribbean	10,258,998	−239,700	−3,408,000 ^†^	296,800
Central America	842,631	−9958 *	−73,740	1035
The Caribbean	595,044	−5629 *	−94,750	6845
South America	6,552,726	−161,900 **	−2,071,000	181,600

^†^ *p*< 0.10, * *p* < 0.05, ** *p* < 0.01, *** *p* < 0.001.

**Table 7 vaccines-12-00238-t007:** Segmented regression of the number of DTP3 vaccinations in American nations.

Nation	Intercept	Year	COVID-19	Interaction
Antigua and Barbuda	1166	−25 *	−242	41
Argentina	747,939	−21,550 *	−440,868	40,473
Bahamas	5272	−147 *	−2177	235
Barbados	2906	−8	−444	19
Belize	73,689	−7	−3833 *	240
Bolivia	246,668	−4856 ^†^	−80,433	6430
Brazil	3,086,377	−101,299 *	−799,678	76,649
Canada	346,374	−1670	−28,913	4283 *
Chile	223,231	−1354	−54,775	6213
Colombia	671,882	277	43,507	−8393
Costa Rica	68,482	−580 ^†^	4127	−856
Cuba	128,843	−2206 *	1376	−726
Dominica	889	1	160	−16
Dominican Republic	183,881	170	−58,735	4679
Ecuador	273,474	−3456	−59,827	3220
El Salvador	116,148	−2777 *	−27,167	1609
Grenada	2040	−12 ^†^	−838 *	42
Guatemala	366,586	−4476	36,154	−5709
Guyana	15,500	49	2205	−220
Haiti	183,283	−3133	−39,333	2490
Honduras	215,078	−3019	−29,014	1396
Jamaica	39,982	−942 *	−8835	955
Mexico	2,170,515	−63,127 *	−1,379,021 ^†^	131,627
Nicaragua	140,894	−108	−4172	−885
Panama	62,938	493	−48,279	4392
Paraguay	123,679	129	48,857	−7328
Peru	546,498	−5221	−372,338 *	34,451 *
Saint Kitts and Nevis	672	−10 *	123	−14
Saint Lucia	2207	−38 ^†^	242	−37
Saint Vincent and the Grenadines	1843	−57 *	−25	1
Suriname	8105	31	−15,211 *	1421 *
Trinidad and Tobago	19,470	−221 *	3238	−372
United States	3,884,049	−44,760 *	−1,093,444 ^†^	107,962 ^†^
Uruguay	49,522	−1632 *	−21,125	2086
Venezuela	549,286	−26,220	−268,317	23,130

^†^ *p* < 0.10, * *p* < 0.05

## Data Availability

Data are available from the United Nations Children’s Fund database.

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
