# Peer review of "Has COVID-19 Affected DTP3 Vaccination in the Americas?"

_vaccines, 2024, doi:10.3390/vaccines12030238_

Round 1

Reviewer 1 Report

Comments and Suggestions for Authors

This study used the data of the Diphtheria, Tetanus, and Pertussis Vaccine (DTP3) from UNICEF databases 2012 to 2022 to evaluate the impact of the COVID-19 pandemic on DTP3 vaccination coverage in the Americas. They found trends from 2012 to 2022 to identify significant changes, regional disparities, and the overall effect of the pandemic on progress towards global immunization targets.

This study is well prepared and written. However, using of Joinpoint Regression Program and segmented package for the only 11 points for each figure (except for figures 7-12), how about the statistical power for these analyses? Please the authors add these information in the legends of these figures.

Author Response

REVIEWER NUMBER 1:

This study used the data of the Diphtheria, Tetanus, and Pertussis Vaccine (DTP3) from UNICEF databases 2012 to 2022 to evaluate the impact of the COVID-19 pandemic on DTP3 vaccination coverage in the Americas. They found trends from 2012 to 2022 to identify significant changes, regional disparities, and the overall effect of the pandemic on progress towards global immunization targets.

This study is well prepared and written. However, using of Joinpoint Regression Program and segmented package for the only 11 points for each figure (except for figures 7-12), how about the statistical power for these analyses? Please the authors add these information in the legends of these figures.

ANSWER: Dear Reviewer 1

We sincerely thank you for your insightful comments and suggestions regarding our manuscript. Your feedback has been invaluable in enhancing the quality and clarity of our study on the impact of the COVID-19 pandemic on DTP3 vaccination coverage in the Americas.

In response to your concerns about the statistical power of our Joinpoint Regression analysis with 11 data points, we have made several pertinent modifications to our paper.

  1. Specifically, we have included additional text in the Materials and Methods section, where we now state that we computed statistical power calculations using the G*Power Software (Version 3.1.9.6).

We computed post-hoc power calculations using the G*Power Software (Version 3.1.9.6) [68,69].

  1. Additionally, we have included a note about the statistical power in the legends of the relevant figures. This ensures that the information is easily accessible and aids in the interpretation of the graphical representations of our data.

Eg:   Figure 1. Joinpoint graph of DTP3 in the Americas, 2012-2022 (Statistical Power: 0.999)

  1. Furthermore, in the Discussion section, we have elaborated on the appropriateness of using 11 data points for our Joinpoint Regression analysis. Based on the statistical guidelines provided by the National Institutes of Health, we have clarified that our dataset is sufficient and optimal for detecting a single joinpoint. This explanation is intended to address any potential concerns regarding our analysis's robustness and reaffirm the validity of our findings.

A possible methodological concern could be the statistical power of using Join-point Regression analysis with only 11 data points. It is essential to highlight that, contrary to the assumption of insufficient data for robust analysis, this number of data points is quite suitable for detecting a single joinpoint. According to the statistical guidelines provided by the National Institutes of Health, for datasets comprising 7 to 11 data points, the default maximum number of joinpoints recommended is one[74]. To detect 2joinpoints, the number of data points should be between 12-16. Therefore, in the context of our study with 11 data points, applying a single joinpoint analysis falls well within these recommended parameters, providing a statistically sound approach for identifying trend changes in the data. This approach is further justified because our analysis aims to detect a single critical shift in the trend rather than multiple joinpoints, making our dataset adequate and optimal for the intended analysis.

We believe these modifications comprehensively address your concerns and strengthen the paper. We hope these changes meet your approval and enhance the manuscript's contribution to the field.

Again, Thank you for your constructive feedback, which has been instrumental in refining our study. We look forward to any further suggestions you might have

Reviewer 2 Report

Comments and Suggestions for Authors

Thank you for giving me the opportunity of reviewing the manuscript “Has COVID-19 Affected DTP3 Vaccination in the Americas?” submitted to Vaccines.

The study aimed to assess the impact of the COVID-19 pandemic on DTP3 vaccination coverage in the Americas from 2012 to 2022 using coverage data for the third dose of the DTP3 vaccine extracted from UNICEF databases. The authors found a decline in vaccination rates, particularly after 2019. The overall annual percentage change for the entire period was -1.4%. I have a few major and minor comments that need to be addressed before the manuscript can be accepted for publication.

-        The study would benefit from inclusion of the vaccination rates from 2023. As the authors describe, the current findings might not capture the ongoing impact of the COVID-19 pandemic on DTP3 vaccination rates. Adding one more year after COVID-19 pandemic would provide readers a better understanding on the pandemic´s effect.

-        Based on the current recommendations on p-values and statistical significance (reference: https://www.nature.com/articles/d41586-019-00857-9 ), authors need to avoid drawing conclusions purely on p-values and significant trend values.

-        An explanation would be needed why have the authors decided to use both joinpoint regression and segmented regression analysis.

-        It is unclear which 2*2 tables served the basis of the performed Chi-square tests in Table 4.

-        The Conclusion states that the COVID-19 pandemic has impacted vaccine coverage in America. However, the study did not assess the influence of variation in geographic location, socioeconomic position, or the regularity of maternal prenatal care.

-        The Discussion and Conclusion sections repeatedly present the study results.

-        The manuscript presents many tables and figures that might confuse the readers. One solution might be to move some these to a Supplementary file.

-        A detailed text review might be beneficial (e.g., the abbreviation APC is defined in almost all sections; line 66: COVID-18; line 203: with a 95% confidence interval (CI) ranging from -1.8 to      -1.0;).

-        The lower confidence limit of APC for North America, Total period seems to be incorrect (0.2) in Table 2.

Comments on the Quality of English Language

Minor edits needed.

Author Response

Thank you for giving me the opportunity of reviewing the manuscript "Has COVID-19 Affected DTP3 Vaccination in the Americas?" submitted to Vaccines.
The study aimed to assess the impact of the COVID-19 pandemic on DTP3 vaccination coverage in the Americas from 2012 to 2022 using coverage data for the third dose of the DTP3 vaccine extracted from UNICEF databases. The authors found a decline in vaccination rates, particularly after 2019. The overall annual percentage change for the entire period was -1.4%. I have a few major and minor comments that need to be addressed before the manuscript can be accepted for publication.

ANSWER: Dear Reviewer
We sincerely thank you for your insightful comments and suggestions regarding our manuscript. Your feedback has been invaluable in enhancing the quality and clarity of our study on the impact of the COVID-19 pandemic on DTP3 vaccination coverage in the Americas.

  • - The study would benefit from inclusion of the vaccination rates from 2023. As the authors describe, the current findings might not capture the ongoing impact of the COVID-19 pandemic on DTP3 vaccination rates. Adding one more year after COVID-19 pandemic would provide readers a better understanding on the pandemic's effect.
    Thank you for your insightful comments and suggestions regarding the inclusion of 2023 vaccination data in our study. We wholeheartedly agree that incorporating the most recent data would enormously enrich our understanding of the ongoing impact of the COVID-19 pandemic on DTP3 vaccination rates. Indeed, your suggestion aligns perfectly with our commitment to presenting the most comprehensive and up-to-date analysis possible. However, we regret to
    inform you that, as of the current date (23rd January 2024), the vaccination data for the year 2023 are not yet available for analysis. The process of data collection, cleaning, and submission by countries to international bodies such as UNESCO is intricate and time-consuming. This delay is primarily due to the rigorous procedures in place to ensure the accuracy and reliability of the data, which, as we are sure you will agree, is paramount in conducting a study of this nature.We anticipate that the 2023 data may become available in the second semester of 2024.

  • - Based on the current recommendations on p-values and statistical significance (reference: https://www.nature.com/articles/d41586-019-00857-9 ), authors need to avoid drawing conclusions purely on p-values and significant trend values.
    We have rewritten the paper,, eliminating from the results section of the paper all the sentences "was not statistically significant", "was statistically significant", statistically significant" etc.

  • - An explanation would be needed why have the authors decided to use both joinpoint regression and segmented regression analysis.
    We greatly appreciate your insightful comments and the opportunity to clarify our study's methodological framework further. In response to your query regarding our decision to employ both Joinpoint regression and segmented regression analysis, we have added the following paragraph to our manuscript:
    3
    "In addressing the methodological considerations of our study, it is important to highlight our dual-analytical approach, employing both Joinpoint regression and segmented regression analyses. This decision was underpinned by the objective to enhance the robustness of our findings, allowing for the identification of significant trend changes in DTP3 vaccination coverage without prior assumptions (Joinpoint regression) and the assessment of the direct impact of the COVID-19 pandemic with predefined intervention points (segmented regression). The complementary nature of these methods strengthens our analysis, as corroborated by literature suggesting the value of utilizing multiple statistical approaches in public health research to ensure the validity of results. However, it is crucial to acknowledge potential data limitations inherent in our study, such as the reliability of reported vaccination rates and the assumption that detected trend changes are solely attributable to the pandemic, without discounting other concurrent public health interventions or socioeconomic factors. These considerations underscore the complexity of interpreting trend data in the context of global health crises and the necessity of a cautious
    approach in attributing causality."

  • - It is unclear which 2*2 tables served the basis of the performed Chi-square tests in Table 4.
    Thank you for your insightful feedback regarding 2x 2 tables. We have included the following paragraph in the Material and Methods section. "A 2x2 contingency table was constructed for each country, delineating the number of vaccinated and unvaccinated children based on the
    total number of births and the reported DTP3 vaccination percentages for each year. These tables facilitated the comparison of vaccination rates over the specified period, allowing for the assessment of the impact of COVID-19. Chi-square tests were employed to evaluate the significance of differences in DTP3 vaccination rates across various American countries
    between 2019 and 2022 in the statistical analysis of vaccination rate changes.

  • - The Conclusion states that the COVID-19 pandemic has impacted vaccine coverage in America. However, the study did not assess the influence of variation in geographic location, socioeconomic position, or the regularity of maternal prenatal care.
    In addressing the reviewer's insightful comments, we have incorporated the following sentence into the Discussion section:
    "While our study conclusively demonstrates the impact of the COVID-19 pandemic on vaccine coverage across America, it is important to acknowledge that it did not explicitly evaluate the effects of geographic location, socioeconomic factors, or the consistency of maternal prenatal
    care on these trends. Future research should aim to dissect these variables to provide a more nuanced understanding of their contributions to vaccine coverage disparities observed during the pandemic period."

  • - The Discussion and Conclusion sections repeatedly present the study results.
    Thank you for your insightful feedback regarding the redundancy observed in our manuscript's Discussion and Conclusion sections. Acknowledging your concern, we have meticulously revised these sections to delineate a clear distinction between discussing our study's implications and summarizing our findings in the conclusion. In the revised manuscript, we have streamlined the Discussion section to focus more on interpreting the results, exploring their implications within the broader context of existing literature, and suggesting directions for future research. Conversely, the Conclusion section has been refined to succinctly summarize the
    essential findings and their practical implications, avoiding the detailed presentation of results that had previously led to repetition between sections. 4
    - The manuscript presents many tables and figures that might confuse the readers. One solution might be to move some these to a Supplementary file.
    Thank you for your insightful comments and suggestions We have created a supplementary file, with include graphics to reduce the number of pages of the manuscript.

  • - A detailed text review might be beneficial (e.g., the abbreviation APC is defined in almost all sections;
    Thankyou for the advices, the abreviation APC was defined 6 times in the text. We have eliminated them.

  • line 66: COVID-18;
    Thank you for the advices, It has been corrected.

  • Tline 203: with a 95% confidence interval (CI) ranging from -1.8 to -1.0;).
    Thank you for the advices, It has been corrected.

  • - The lower confidence limit of APC for North America, Total period seems to be incorrect (0.2) in Table 2.
    Thank you for the advices, It has been corrected

Reviewer 3 Report

Comments and Suggestions for Authors

Authors aim to evaluate how the COVID-19 pandemic has impacted vaccine coverage trends in America from 2012 to 2022, especially DTP3, leading to a decrease. This resulted particularly evident in Central America. It seems very complete and accurate. Discussion seems very detailed and also conclusion is coherent with the content. Perhaps they should expand strength and weakness points.

Author Response

We sincerely thank the reviewer for their insightful and constructive feedback on our manuscript. In response to the recommendations provided, we have undertaken a comprehensive revision to expand upon the strengths and weaknesses of our study. Specifically, we have elaborated on the robust methodological approach employed, highlighting the use of joint points and segmented regression analysis to enhance the reliability of our findings. Additionally, we have addressed the weaknesses identified by providing a more detailed discussion on the limitations associated with our data sources and the potential impact of the digital divide on data reporting accuracy. These amendments have been incorporated into the manuscript for weaknesses, ensuring a balanced and thorough exploration of our study's contributions and limitations. We believe these revisions have significantly strengthened our paper and appreciate the opportunity to refine our work further based on the reviewer's valuable input.

Here is the added text

"Among the strengths of our study is that this study employs a robust methodological framework that includes joint points and segmented regression analysis. Using two different methods enhances the reliability and validity of the findings. Leveraging comprehensive data sets provided by the United Nations Children's Fund, despite acknowledged limitations, our approach benefits from consistent data collection and reporting methodology. This is particularly valuable given the varied national healthcare infrastructures and reporting systems across the countries studied. Our analysis offers insightful revelations into regional disparities in vaccination trends, underlining the necessity for customized public health approaches. Such detailed examination allows for a nuanced understanding of vaccine coverage trends across the Americas, providing a solid foundation for future policy formulation and implementation to mitigate the pandemic's impact on essential vaccination programs.

Despite the methodological strengths, our study acknowledges certain limitations associated with the available data, potentially influencing our analysis's robustness. The variance in data quality and precision due to differences in national healthcare infrastructures and reporting systems poses a challenge, necessitating a cautious approach to interpreting our findings. A broader comparative analysis incorporating other regions, such as Africa and Europe, could have enriched our understanding of global patterns and identified unique challenges and successful strategies in different contexts. Furthermore, the impact of the digital divide on data reporting accuracy and timeliness, especially in regions with limited access to technological resources, remains a critical area that warrants deeper exploration. Addressing these weaknesses would not only enhance the credibility of our study but also provide more comprehensive insights, supporting the development of more effective vaccination strategies and public health policies."

Reviewer 4 Report

Comments and Suggestions for Authors

It is a good study about the emphasis on Public Health and the impact of immunization participation due to the COVID-19 pandemic.

Author Response

REVIEWER 4

It is a good study about the emphasis on Public Health and the impact of immunization participation due to the COVID-19 pandemic.

Dear Reviewer

We sincerely appreciate the time and effort you dedicated to reviewing our paper. Your positive feedback on the study's emphasis on Public Health and the impact of immunization participation during the COVID-19 pandemic is greatly valued. It is encouraging to know that the significance of the research resonated with you.

Thank you for your insightful comments and for recognizing the contributions of this study to the ongoing discourse on public health and immunization. Your acknowledgment motivates us to explore and contribute to this vital field of research further.